# Catastrophic Floods in Large River Basins: Surface Water and Groundwater Interaction under Dynamic Complex Natural Processes–Forecasting and Presentation of Flood Consequences

Tatiana Trifonova [1], Mileta Arakelian [2], Dmitriy Bukharov [3], Sergei Abrakhin [3], Svetlana Abrakhina [3] and Sergei Arakelian [3],*

1   Department of Soil Geography, Lomonosov Moscow State University, 119991 Moscow, Russia; tatrifon@mail.ru
2   Department of Theoretical Physics, Yerevan State University, Yerevan 0025, Armenia; marakelyan@ysu.am
3   Department of Physics and Applied Mathematics, Stoletovs Vladimir State University, 600000 Vladimir, Russia; buharovdn@gmail.com (D.B.); abrahin_s@vlsu.ru (S.A.); abrahina_s@mail.ru (S.A.)
*   Correspondence: arak@vlsu.ru

**Abstract:** A unique approach has been developed for explaining and forecasting the processes of flood and/or mudflow (debris) formation and their spread along riverbeds in mountainous areas, caused by flash increases in the water masses involved (considerably increasing in their expected level because of precipitation intensity) due to groundwater contributions. Three-dimensional crack-nets within the confines of unified rivershed basins in mountain massifs are a natural transportation system (as determined by some dynamic external stress factors) for groundwater, owing to hydrostatic/hydrodynamic pressure distribution, varied due to different reasons (e.g., earthquakes). This process reveals a wave nature characterized by signs of obvious self-organization, and can be described via the soliton model in nonlinear hydrodynamics on the surface propagation after a local exit of groundwater as the trigger type. This approach (and related concepts) might result in a more reliable forecasting and early warning system in case of natural water hazards/disasters, taking into account a groundwater-dominant role in some cases.

**Keywords:** catastrophic floods; rivershed basin; surface and groundwater interaction; statistical analysis; 3D crack-net structure; seismic processes impact

## 1. Introduction

The backgrounds and basic principles of catastrophic floods are usually reduced to a standard view about heavy rainfall [1,2] but without real forecasting or preliminary measuring and monitoring of key factors. Thus, many problems still exist, and the knowledge level concerning catastrophic mudflows/debris and floods in mountainous conditions is still insufficient (see, e.g., [3–5]).

Indeed, as a presentation example of [6], flood causes in Europe (2013) are traditionally quite obvious, although disastrous flooding is usually caused by a set of reasons. The leading factor for such periodically rising water events is heavy rainstorms (up to 4–6 inch/day) being far too heavy in Europe for the considered areas [7]. In fact, the two-month precipitation rate fell in a day (15 July 2021). However, today we do not have simple models that would allow us to analyze (see also [4]) and, moreover, predict such extreme events, especially for fairly rapid flooding/debris in mountainous conditions in rough terrain. After all, the standard position is associated with heavy rains, even without taking into account the specific terrain of the territory and the high probability of extra water flow through the river basin system in general.

However, our main idea, as discussed in this paper, deals with floods occurring as a result of several factors of influence. Namely, the interaction between the surface (here

meaning all water objects of any type in the considered areas—lakes, artificial reservoirs and river networks) and groundwater (from different water horizons) is the vital factor in certain cases of disastrous floods, especially in major river basins, even during heavy precipitation periods lasting several days.

Moreover, some strange indicators appear when we try to analyze the flooding process. Here, those uncertainties are presented in the form of four questions as a background and basis for the article's motives (perhaps this is not quite a standard presentation, but it is reasonable for a better understanding of the problem). Such problematic issues can be listed as follows:

**Question 1.** Is there an obvious data discrepancy between the estimation of rainfall levels in an area and an observable increase in water discharge in a riverbed, and/or is this due to the difficulty of making calculations and measurements in a selected territory with a complex landscape?

In fact, we have assessed the water balance of floods (see Figures 1 and 2) based on available official data (summarized through the region) [7] in two examples (the percentage differences in the discrepancy were calculated arbitrarily based on the maximums of the water masses observed). First, the 2015 Louisiana flood (USA), near the City of Shreveport: the accumulated water volume mass was ~3.3·$10^9$ m$^3$, but the observed water volume mass was ~11.0·$10^9$ m$^3$. Thus, the relative difference between the maximal values of the accumulated and observed water masses was more than three times. Second, the same issue can be found in the example of the 2015 Assam flood (India): the accumulated water volume mass was ~26.5·$10^9$ m$^3$, but the observed water volume mass was ~31.4·$10^9$ m$^3$. Thus, the relative difference between the maximal values of accumulated and observed water masses was about ≈15% (cf. [8]).

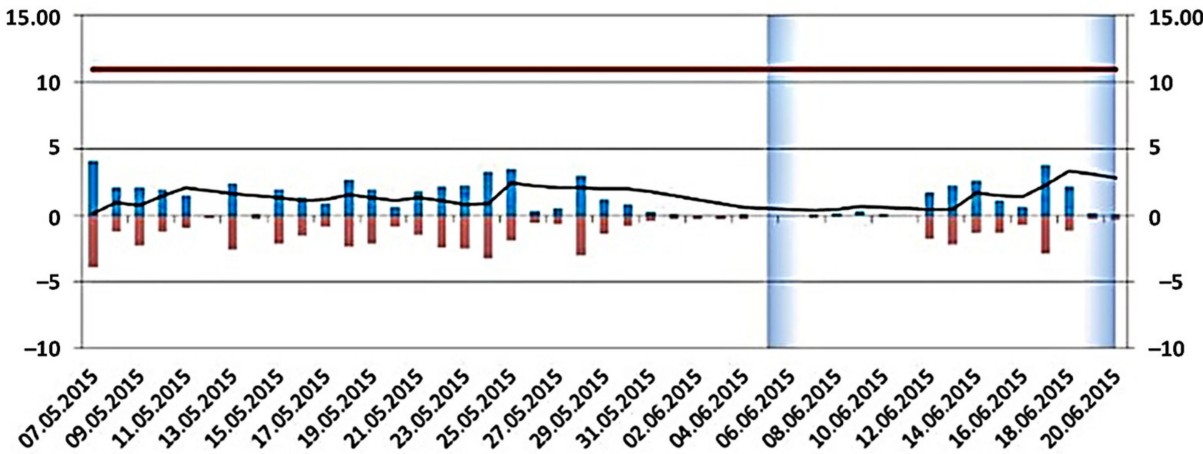

**Figure 1.** Water balance estimation for the example of the 2015 Louisiana flood. Blue bars refer to the whole volume of daily precipitation in the whole basin (summarized through regions) in units $10^9$ m$^3$; red bars–the whole volume of daily evaporation + permeation in the whole basin (summarized throughout the region), $10^9$ m$^3$; black line–the whole volume of accumulated water mass in the whole basin (summarized through regions), $10^9$ m$^3$; red line–the maximum of observed water mass, $10^9$ m$^3$. On the vertical axis–the water level ($10^9$ m$^3$). On the horizontal axis–measurement days (date). Positive values–excess water mass compared to normal conditions, negative values–decrease compared to normal conditions.

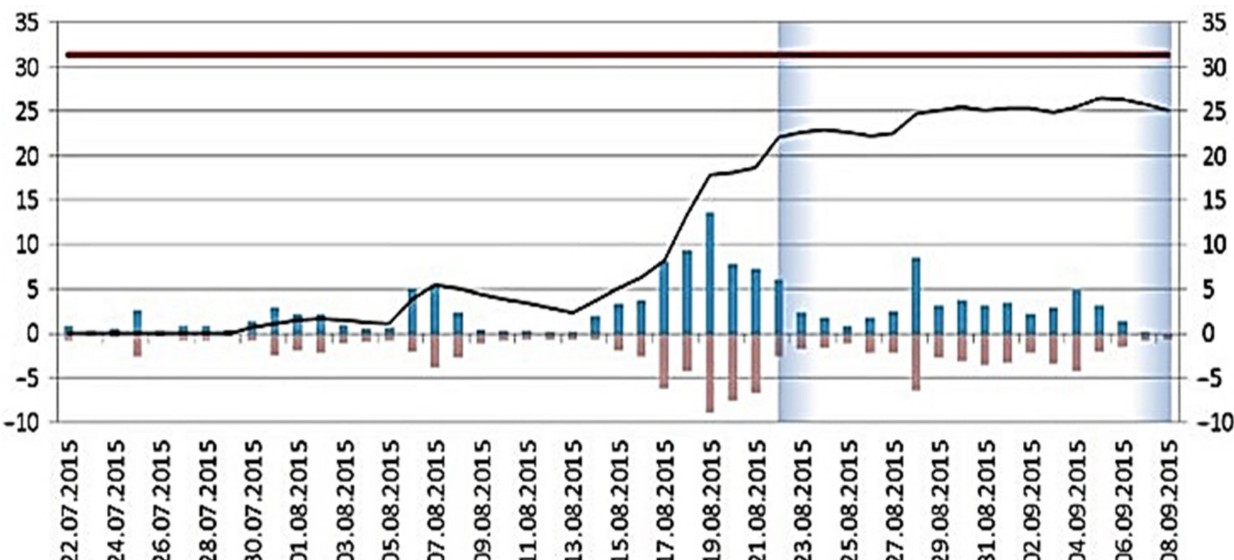

**Figure 2.** Water balance estimation for the example of the 2015 Assam flood. Blue bars refer to the whole volume of daily precipitation in the whole basin (summarized through regions), in units $10^9$ m$^3$; red bars–the whole volume of daily evaporation + permeation in the whole basin (summarized throughout the region), $10^9$ m$^3$; black line–the whole volume of accumulated water mass in the whole basin (summarized through regions), $10^9$ m$^3$; red line–the maximum of observed water mass, $10^9$ m$^3$. On the vertical axis–the water level ($10^9$ m$^3$). On the horizontal axis–measurement days (date).

All detailed databases on the subject can be introduced by event analysis using [7,9,10].

**Question 2.** In the previous question (1) we indicated discrepancies by estimations only as fact. However, we now suggest a possible reason for such discrepancies, caused by the release of groundwater upon the surface. Indeed, why does extra water mass appear during such events? Is it an accumulation effect observed due to the complex specificity of landscape, with accumulation somehow taking place in only one river bed and, moreover, water stagnating over a long time? Highly likely, this happens due to groundwater contribution in localized areas. This fact can result in long distances and durations in these events, e.g., the catastrophic floods in Louisiana (USA), 2015.

In this aspect, we can compare two databases [10,11].

In 2015, in the Red River basin near Lawton, catastrophic flooding occurred: 16 June: 2.0 inch/day; 18 June: 3.33 inch/day; 20 June: 3.0 inch/day.

However, in 2016 in Baton Rouge, flooding did not occur: 12 August: 11.24 inch/day; Lafayette, 12 August: 10.39 inch/day; 13 August: 10.40 inch/day.

In addition, during flooding in late May and June 2013 in Western Europe (in the river basins of the Danube, Elbe, Rhine, etc.) the water level rose by 7–13 m, and two-month rainfall fell over only two or four days: 4 June 2013: Austria, 170–220 mm; 6 June 2013: Germany, 150–180 mm, the water volume was 23 km$^3$ [6,10,12]. However, in contrast, in Moscow and the Moscow region (e.g., the town of Kashira), practically at the same time and for a similar landscape, in September 2013, more than 180 mm (exceeding the average level by three times), and 277 mm (exceeding the average level by five times) fell daily, respectively, but no catastrophic flooding occurred [12]. This means that, under the concept that special conditions are required for groundwater to release to the surface, catastrophic flooding will result.

To support this idea, we simulated the instant collapse (explosion) of an artificial reservoir dam with a water mass parameter of 4.5 million m$^3$, square: 5 km $\times$ 0.5 km, discharge from 80 km$^2$ of the small river of Sodyshka, near the city of Vladimir (Russia) [13]. The process of water flow for the event was very fast (a few hours) and resulted in local flood areas around the river bed (practically, the water level is not above the river bed table over the river channel due to historic natural development).

Thus, torrential rain is probably an obligatory but not sufficient condition. Moreover, sometimes a strange phenomenon is observed in areas (especially in wooded areas) after catastrophic flooding: fires burst there within several months/the next year. This fact might be explained by the depletion of groundwater resources in the area. The impact of early flooding on accidental fires in the near future can be demonstrated with the event at the Amur River (Russia): catastrophic flooding from August–September 2013, and then powerful fires in April 2014 [13,14].

In addition, we have noted that incessant heavy rain in 22 July 2021 caused the collapse of a Trans-Siberian Railway bridge, Russia [15]. Judging by the footage from the scene, it can be concluded that bridge supports were washed away as a result of strong currents. Due to this fact, it is hard to believe that only surface flow is to blame. Instead, it is probably due to the impact of the groundwater table, in which the powerful bridge supports, located in the depths, were damaged, and a strong variation in the groundwater state might also result in such destruction.

**Question 3.** What are groundwater's transport routes up to the surface? Are natural, permanent water sources from underground horizons (springs, geysers, grottos, etc.) providing the directions?

The answer lies in the fact that 3D fractures in geological structures and rocks within underlying surfaces, including dry riverbeds, have crack topology infrastructures that go in many directions, including in deep layers. [2,5,13,16,17].

Indicative in this regard, we turn to the long-term, catastrophic flooding of 2013 (July–September) on the Amur River (Russia) [14]. Despite heavy rains (about 50 mm per day) that covered large areas, including both the main channel and its numerous tributaries, the flooding itself spread only around the main channel (see Figure 3). In this case, the increase in water consumption started at a level of ~20·$10^3$ m$^3$/s and reached a level of ~46·$10^3$ m$^3$/s. A possible explanation for such a smooth, long-term process is: generally, groundwater self-discharging to the surface can take place only in a fixed area in a main channel for a spatially distributed system like a river basin (e.g., this major river), delocalized over a large region. Indeed, traditionally, flooding should spread both along the main channel and along tributaries (usually embracing large areas where it rains heavily; see Figure 3c), and it also should not last long throughout the territory. However, this was not the case for the considered event: even along geographically close tributaries, the situation was different, even when taking into account dry river beds—see Figure 3a,b. To support this point of view, we can predict many events when a small river's discharge becomes comparable to the discharge of major world rivers due to the fantastic localization of its water mass in a small, isolated channel, and even with strongly dissected relief (see, e.g., [18,19]).

A discussion of the universal concept of the groundwater's role in catastrophic floods should also include a statement dealing with the total global groundwater resources unified in the river basins of different/neighboring rivers, especially those lying close to the Earth's surface [11].In the considered case, a principal consequence of the common groundwater resources of different major rivers is the Lena riverbed (Russia) shallowing, caused by catastrophic flooding on the Amur River [10,14,19]. Apparently, this phenomenon was associated with the temporary depletion of the river's total groundwater resources before a subsequent restoration over time by various mechanism. Here, we are talking about the connection of the groundwater basins of different rivers, even major ones; i.e., in this case, they have different surface discharge systems, isolated by topography, but do not necessarily have different underground resources; they might be unified as a shared underground network.

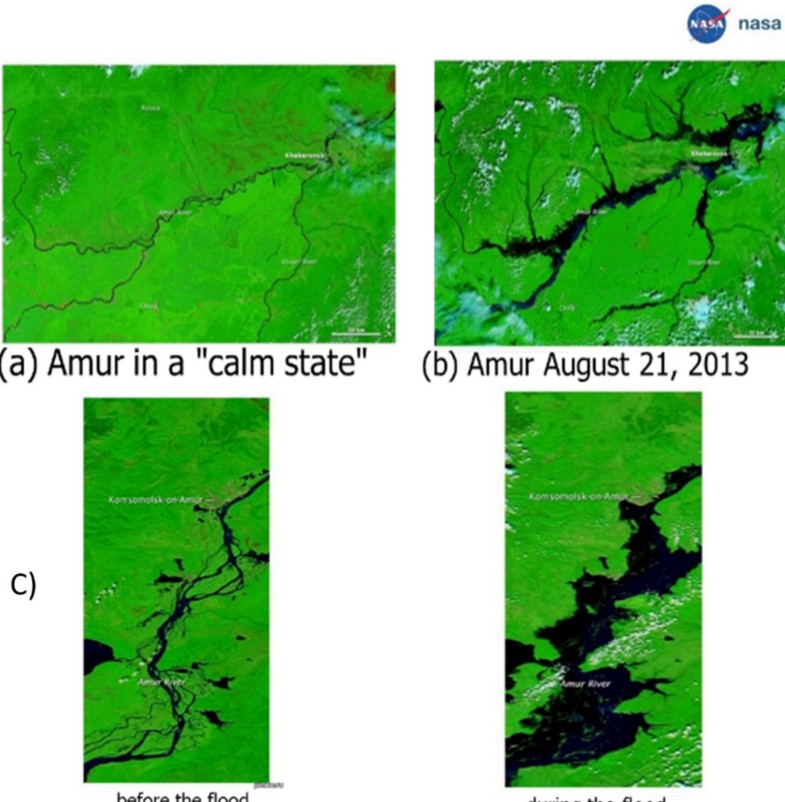

**Figure 3.** Collection of space images (NASA). Flood inundation areas versus the hydrological situation during calm times: (**a**) before the flood; (**b**) during the flood; (**c**) in the Komsomolsk-on-Amur city area (Russia). The pictures were simultaneously obtained for some surface water states, but show irregular distribution for opening the transport waterways for groundwater over a large area along the "activated" parts of the 3D river-drainage system.

**Question 4.** Why does the preliminary, stable, steady-state process of the unified water system of the rivershed basin become unstable in its dynamics regarding surface water mass flow? Is this the naturally and externally induced lifecycle of the water system, and/or is a variation in the soil moisture taking place [4,5,17,20]?

Obviously, no one doubts the connection between groundwater and rivers on the surface due to the important, ordinary groundwater contribution of the well-known hydrological processes [5,20]. However, we are talking about the fact that equilibrium in the dynamic state is disturbed with the extreme access of groundwater on the surface under certain conditions during catastrophic floods. The search for these reasons is the subject of consideration in our article.

In this case, instability variation in a 3D river network system might occur due to changes in the 3D map of both crack-nest topology and pressure distribution in underground horizons, as objects associated with water tables, due to subsurface, external causes. This happens not only because of rain but also due to openings forming new, underground channels (previously blocked) due to increased pressure on depth channels from surface water objects, like lakes and artificial reservoirs (both up and down a river's flood-area localization) caused by extra rainwater mass. However, the principal point concerns the impact of microseismic events and earthquakes on the development of trigger processes.

Thus, the traditional approach implies that surface runoff is only the endpoint of flood development (see, e.g., [1,2,5]):

(1)    All water is formed from precipitation according to the local terrain;
(2)    Surface water is considered separately from groundwater during any event.

However, according to our approach, all water systems are closely interconnected and varied, especially during catastrophic floods, so none of them are the endpoint. In this aspect, surface water, groundwater and geological structure function as a unified system under the dynamic processes of their lifecycles, especially due to the impact of the external factors.

In this paper, an approach for pre-forecasting is discussed for explaining and forecasting the flood and/or mudflow (debris) formation processes, as well as the nonlinear hydrodynamic phenomenon of spreading out water mass over river beds in mountain conditions. The fact is that, usually, when it comes to the floods, only the precipitation level is analyzed as a universal, key parameter. Our entire article questions this thesis concerning certain, specific cases and the processes caused by flash increases in the water masses due to groundwater (considerably increasing the expected level thanks solely to the precipitation's intensity).

In this case, a 3D crack-net, within the confines of a unified rivershed basin in a mountain massif [2,5,16,20,21], is a natural transportation system for groundwater, varied by dynamic stress from external factors. Thus, a map of hydrostatic/hydrodynamic pressure distribution is a key point in understanding groundwater horizons in different states and flow, including the impact of earthquakes of any magnitude [13,21].

At the end of the introduction, we consider it reasonable to pre-summarize the principal aspects of our concept, developed in the next sections of the presented article.

According to our approach, a riverbed is not just a surface formation but a part of a 3D water structure. The basic principle of this concept is that river/stream fractures are laid down and propagate in the rock as a result of accumulated stress relaxation. The cracks formed in the rock extend not only along the surface, but also into great depths (several hundred meters and/or even more). Groundwater is pulled into the crack zone; the mechanism of this 3D process is connected to the action of internal (deep) pressure and capillary forces (in the latter case, the water flow may spread at extremely high velocity (see, e.g., [2])).

As a result, in the zone of such deeply fractured riverbeds, a directional rise of groundwater up to the surface occurs. These waters are essential for river basin formation and permanently (year-round) influence the functionality of river water systems [8,13,16].

As to surface flow, this is another, non-permanent component of the water balance, which mainly depends on climatic conditions.

The paper is organized as follows:

In Section 2, we discuss the methods behind our basic concept, as well as both the database involved and dynamic models for earthquake impact on floods due to 3D crack network reconstruction.

The results of our study are presented in Section 3. We consider a short statistical analysis for some localized 3D river basin areas using several parameters: river discharge, precipitation level a artesian water level in wells. The study indicates the groundwater state and proposes the required frequency of parameters measured in time. Additionally, possible schemes of earthquake impact on several real floods are discussed.

In Section 4 (Discussion), the complex analysis results are considered, taking into account the basic principles of the possible influence of tectonics on groundwater-transportation-system function.

In conclusion (Section 5), we briefly discuss the practical verification of the risks for catastrophic flood development according to the proposed approach.

Appendix A includes several additional, objective databases useful for understanding the basic concepts of our approach.

## 2. Methods

Several objective databases and their possible interpretations are discussed below, helping to understand our approach. This is a non-standard concept, and in our opinion, is a plausible hypothesis with a number of simple, preliminary demonstrations of several

specific examples, taking into account the fundamental question of why the state of groundwater and its transportation routes to the surface suddenly change dramatically at certain times, even though, up until then, everything was in a dynamic equilibrium state. It is difficult to demand a complete, general proof across all the available, numerous databases on the problem from an initial, accentuated statement, but we are trying to demonstrate reasonable tendencies. In addition, in our conclusion (Section 5) and in Appendix A, we have provided some useful data on the subject.

### 2.1. Basic Concept of General Approach

Now we discuss the basic principles of the three phenomena in competition, varied in different time and space scales:

(1)     Precipitation in a specific, selected area and its level estimation in quantitative parameters;
(2)     Discharge and water flow processes along the river bed, and related measurements that were carried out;
(3)     Groundwater distribution with regard to its volume and lifecycle by monitoring their state at the time.

The key point of the problem is water-balance estimation during the event. However, with regard to the standard estimation procedure within the general model of the water accumulation process (see Figure 4—cf. [1,2,4,5]), many questions and uncertainties are discussed.

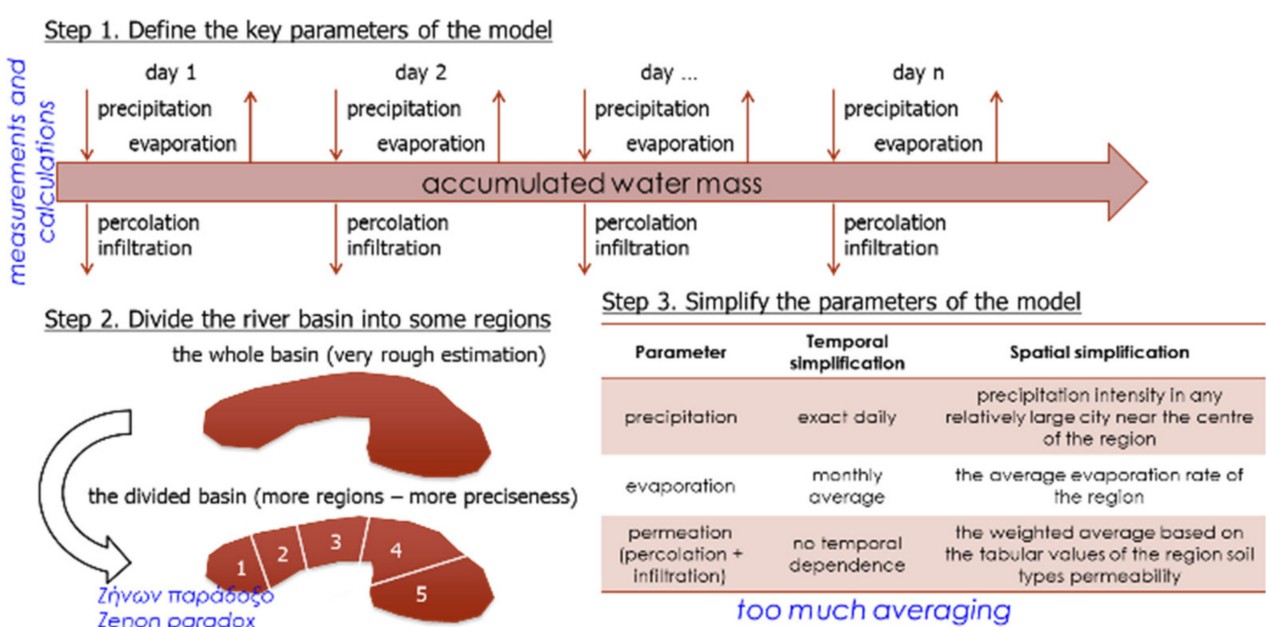

**Figure 4.** General schematic model of water accumulation and water-balance estimation.

At any rate, we are aware of certain fundamental information about rock mechanics and river modeling [20,22], but, nevertheless, we can simplify the analysis procedure.

In general, it is difficult to describe such rapid dynamic processes embracing so many factors, developing in real-time in fixed, stationary intervals (analogue to the well-known problem of Zeno's paradox [23]).

We have used a more fundamental point in our research methods, including an approach stating that, to analyze a flood's development, it is necessary to take into account the influence of various factors dealing with the sudden change in the state of the 3D system of the river basin in dynamic regime, primarily caused by earthquakes. Figure 5 schematically demonstrates this; i.e., the standard (a) and proposed procedure (b).

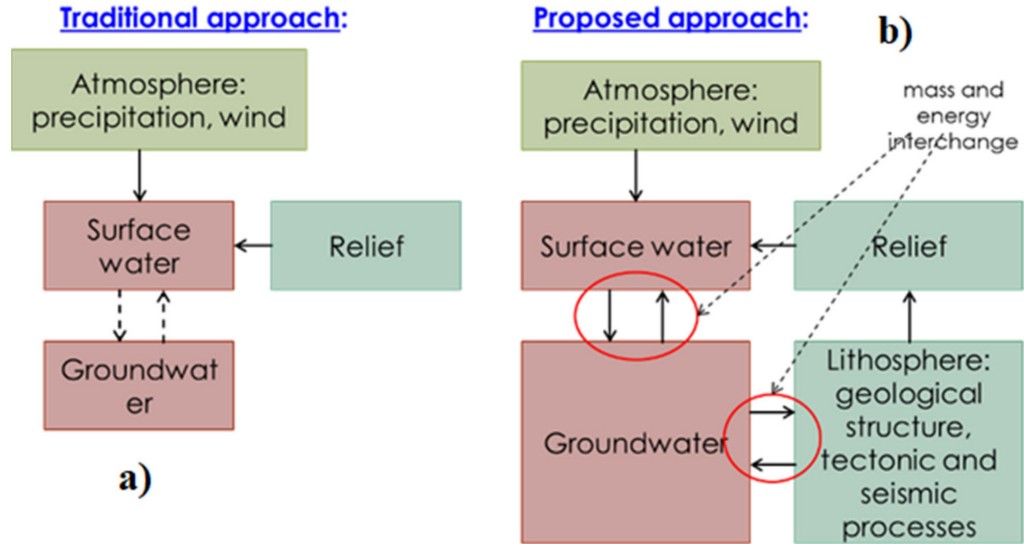

**Figure 5.** The key parts of a river basin system's functions: (**a**) traditional view; (**b**) our proposed model.

Water flow variations under conditions of reconstruction in a 3D crack-net are schematically illustrated in Figures 6 and 7.

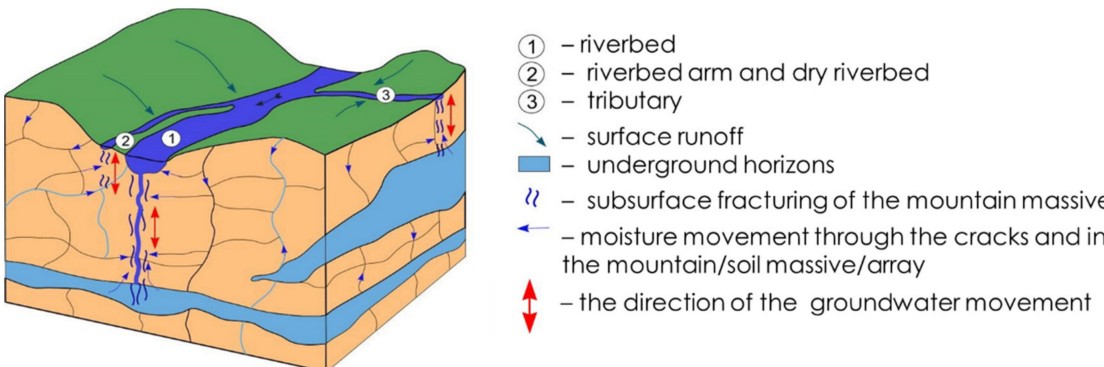

**Figure 6.** River functioning in a "normal" state.

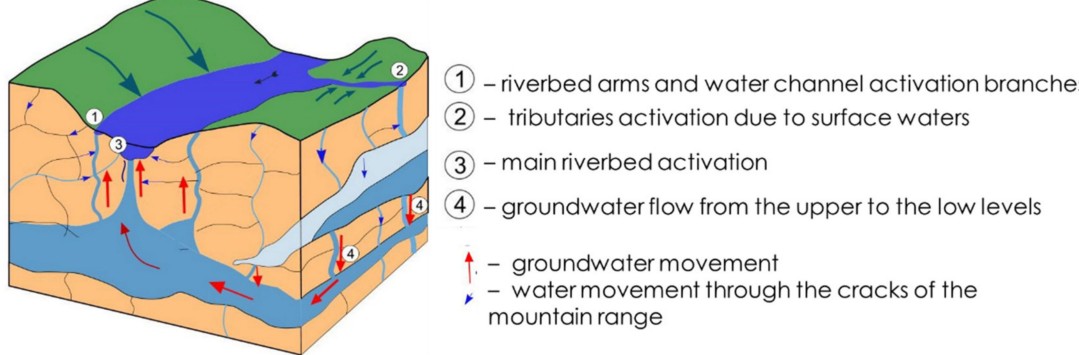

**Figure 7.** Reconstruction of the mountain's massive fracturing and the river's underground water supply.

As to the possible impacts on groundwater exits to the surface due to the configuration/reconstruction of crack-nets (cf. [17,21]), the following earthquakes occurred in Russia with a natural time delay that could have affected the above-discussed Amur River event (see Figure 3):

5.0, 4.4, 2.9 magnitudes, Sakhalin Island (4, 7, 9 July, respectively, 2013);

5.9 magnitude, Kamchatka (17–18 July 2013);

Volcanic activity in Kamchatka, Klyuchevskaya Sopka, Shiveluch (Summer, 2013);

Japan: 6.9 magnitude, Pacific coast of Japan (4 September 2013).

Further, after groundwater releases to the surface, localized in a certain place, the process is characterized by a wave type with obvious signs of self-organization, and it can be described (see, e.g., [24]) within the soliton model of nonlinear hydrodynamics when the groundwater propagates over the surface after the local discharge exit as a trigger unit.

This approach (and the corresponding concept) may result in a more reasonable preforecast and early warning system for natural water hazards/disasters, taking into account groundwater's dominant role in specific areas (see Section 3: Results and Section 5: Conclusions).

### 2.2. Database and Complex Analysis

Now we take into account a reasonable factor within the confines of the basic principles of tectonic impact on groundwater functionality.

First, we highlight the data collection undertaken for the subject under consideration; i.e., we collected data concerning earthquakes and floods.

Second, during this data collection and analysis, it was necessary to solve a kind of clustering task, which was determined by several factors of different types of information:

(1)    Only disastrous/historical events (for observing the extremes of considered parameters);
(2)    No coastal regions (excluding tsunamis);
(3)    No seasonal events (excluding freshets);
(4)    Acceptable spatial and temporal lags—not more than a month.

As an example of the summarized results, we display the data in Table 1 (according to [21,25–27]).

**Table 1.** Data on several earthquakes and floods.

| Earthquake Location | Geographical Coordinates of Epicenter | Date | Magnitude | Depth of Hypocenter | Flood Location | Flooding Period | River Basin |
|---|---|---|---|---|---|---|---|
| Montenegro | 43.15° N 18.86° E | 21 May 2013 22:55 | 4.5 | 10 km | | | |
| Bosnia and Herzegovina | 43.81° N 17.05° E | 20 May 2013 9:24 | 4.0 | 10 km | Germany Czech Republic Austria | May–June 2013 | Danube Elbe |
| Algeria | 36.85° N 5.10° E | 19 May 2013 9:07 | 5.1 | 10 km | | | |
| Muğla Province, Turkey | 36.96° N 28.49° E | 16 May 2013 3:02 | 5.0 | 10 km | | | |
| Texas, USA | 32.03° N 94.42° W | 2 September 2013 23:51 | 4.5 | 10 km | | | |
| Mexico | 27.77° N 105.68° W | 28 August 2013 20:29 | 4.3 | 10 km | Colorado, USA | September 2013 | Boulder |
| California, USA | 39.80° N 120.13° W | 27 August 2013 0:51 | 4.2 | 10 km | | | |
| Kansas, USA | 37.52° N 98.74° W | 23 May 2015 18:44 | 4.0 | 10 km | Louisiana, USA | June 2015 | Red River |
| Kyrgyzstan | 41.93° N 76.80° E | 28 April 2017 5:01 | 4.7 | 10 km | | | |
| Xinjiang, China | 37.88° N 78.13° E | 20 April 2017 3:39 | 4.6 | 10 km | | | |
| Afghanistan | 36.51° N 70.93° E<br>36.70° N 71.51° E<br>36.42° N 69.17° E | 17 April 2017 23:04<br>4 April 2017 4:48<br>2 April 2017 2:48 | 5.0<br>4.8<br>4.8 | 184 km<br>167 km<br>46 km | Kazakhstan Tyumen oblast, Russia | April–May 2017 | Ishim |
| Tajikistan | 37.76° N 72.19° E | 10 April 2017 6:57 | 4.8 | 110 km | | | |
| Iran | 35.73° N 60.42° E<br>31.23° N 60.43° E | 5 April 2017 6:09<br>4 April 2017 0:12 | 6.1<br>4.5 | 15 km<br>10 km | | | |
| Kazakhstan | 47.19° N 85.06° E | 4 April 2017 15:07 | 5.1 | 10 km | | | |
| Mexico | 19.62° N 95.90° W<br>17.21° N 99.54° W<br>17.60° N 100.97° W<br>17.87° N 94.40° W<br>16.79° N 98.26° W<br>16.26° N 98.75° W | 15 February 2017 9:56<br>13 February 2017 7:29<br>2 February 2017 0:52<br>25 January 2017 20:54<br>12 January 2017 10:26<br>7 January 2017 6:16 | 4.4<br>4.7<br>4.7<br>4.9<br>5.0<br>4.6 | 32 km<br>34 km<br>23 km<br>179 km<br>39 km<br>10 km | California, USA | February–June 2017 | Sacramento |
| Vancouver Island, Canada | 49.38° N 129.30° W<br>49.92° N 127.60° W<br>50.22° N 129.95° W | 12 February 2017 3:47<br>31 January 2017 1:38<br>6 January 2017 15:49 | 4.7<br>4.1<br>5.3 | 10 km<br>10 km<br>10 km | | | |

Our preliminary analysis from different sources (cf. [13,17,21,26]), based on data according to the International Seismological Centre, 1990–2019, with an average of every 4 years for many events (more than two dozen, 2010–2017), allowed us to determine the likeliest parameters of the greatest risks for an earthquake's impact on catastrophic floods (see Figure 8): depth of hypocenter, ~10 km; point of epicenter on the Earth's surface, ~VII; magnitude ~5, i.e., by energy, ~$10^{12}$ Joules.

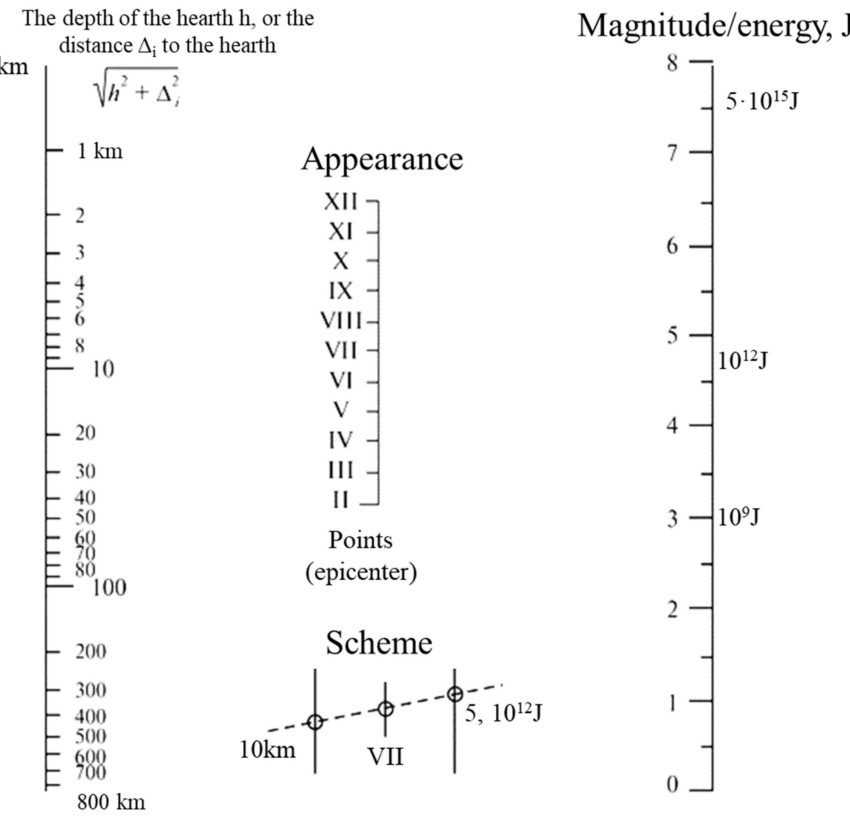

**Figure 8.** Highest probability for catastrophic water events (by analysis from different sources): magnitudes, intensity (in points/earth scores), and focal depth (hypocenters) that cause catastrophic floods when an earthquake occurs. The data shown as a scheme is the max-risk for the event (all the above 3 factors come together).

### 2.3. Dynamic Models and Reconstruction of 3D-Crack-Net under External Factors

A more complicated and universal dynamic model, i.e., the propagation effect of seismic waves (from their different sources) in rock fracturing, may be established by the SIR (Susceptible Infected Removed) model (cf. [27]): agents interacting in various physical states. In this process, the model implies 3 possible agent states: "Vulnerable"— $S(t)$ (Susceptible), ready to accept the sign/state; "Unresponsive"—$R(t)$ (Removed), will not perceive the sign; "Infected"—$I(t)$ (Infected), the agent has already successfully accepted the sign and is ready to spread it. This approach also uses two parameters characterizing the model process—the propagation rate of the trait ($\beta$) and the rate of "Immunization" ($\gamma$), which can be interpreted as the saturation rate of the trait.

The equations for this case are [28,29]:

$$\frac{dS(t)}{dt} = -\beta S(t)I(t),$$

$$\frac{dI(t)}{dt} = \beta S(t)I(t) - \gamma I(t),$$

$$\frac{dR(t)}{dt} = \gamma I(t),$$

with initial conditions $S(0) = S_0 > 0$, $I(0) = I_0 > 0$, $R(0) = R_0 > 0$.

The analysis results are shown in arbitrary units in Figures 9a–j and 10a–f—the explanations are given in Figures.

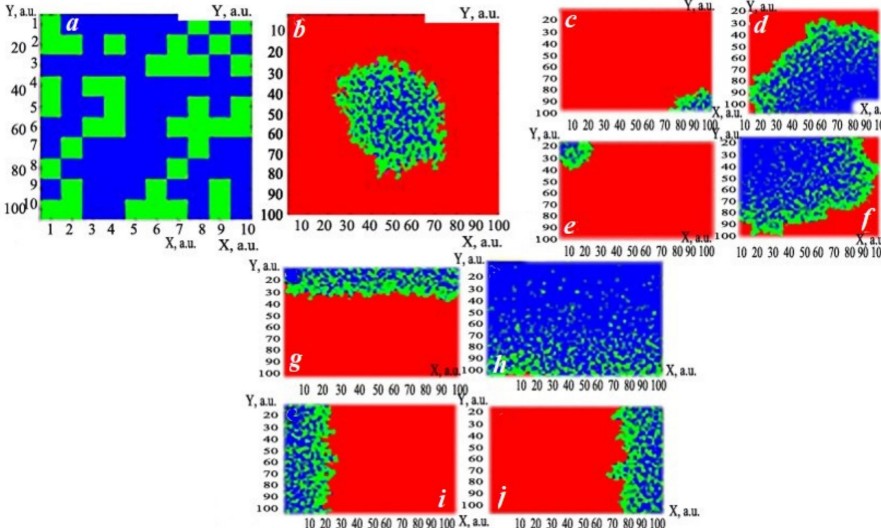

**Figure 9.** SIR model for seismic wave propagation from sources: $S(t) + I(t) + R(t) = const = N$ (from the number of objects N). Solutions (in arbitrary units) of the trait propagation model by cellular automaton method for β = 0.029, γ = 0.01, T = 100: (**a**) N = 100, S(0) = 10; (**b**) N = 10000, S(0) = 10. Different colors indicate the cell states—from 0 to 2; for β = 0.029, γ = 0.01, N = 10000: (**c**) T = 100, lower-right corner; (**d**) T = 500, lower-right corner; (**e**) T = 100, upper-left corner; (**f**) T = 500, upper-left corner; for β = 0.029, γ = 0.01, N = 10,000: (**g**) T = 100, the upper limit of the computation domain; (**h**) T = 500, the lower limit; (**i**) T = 100, the left border; (**j**) T = 100, the right border. Here, T stands for the relative number of steps in time and specifies the distribution in the uniform grid with step h.

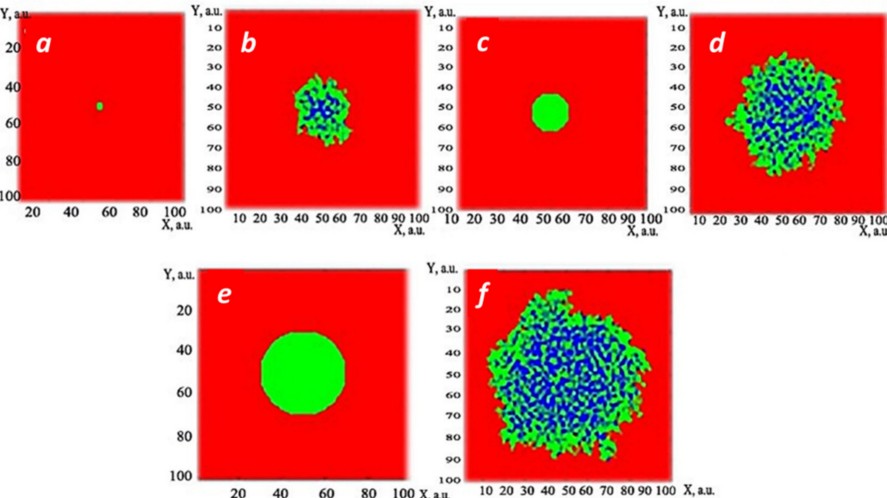

**Figure 10.** Seismic process propagation from a single isolated source (radius r). Initial conditions and solution–image for the propagation region of the studied state (in arbitrary units): (**a**) initial conditions r = 2, T = 4; (**b**) corresponding solution, but already for T = 100; (**c**) initial conditions r = 10, T = 4; (**d**) decision, but for T = 100; (**e**) initial conditions r = 20; T = 4; (**f**) decision, but for T = 100, where T is also the relative number of steps in time and specifies the partition on the uniform grid with step h.

The integration of the obtained images helps systemize flood risk areas under the influence of tectonic processes for both boundary propagating events and singular isolated sources.

Interpretation of the obtained images helps us to systemize the risk areas for flooding under the influence of tectonic processes for both boundary propagating events and singular sources.

Since groundwater transport routes are very sensitive to external influence, it is necessary to analyze the reconstruction of the 3D crack-net due to external factors [13,16,21], cf. Figures 6 and 7.

However, now the propagation anisotropy (due to inhomogeneous medium) should be taken into account. We carried out a simulation modeling for that using a computer program for modeling pressure maps in groundwater within fractured rocks. The main points for this procedure included the following items (cf. [30]):

(1) The basis was crack fractal modeling in the rock structure;
(2) Cracks were superimposed on the earth surface profile where points of crack emergence on the surface were formed.
(3) It was assumed that the entire crack network was filled with water;
(4) The pressure in the head fracture was set, and the computer algorithm calculated what pressures would be at the emergence points of different surface exits;
(5) Excluded cracks not coming to the surface and could create a tension zone inside the rocks.

The results obtained by this procedure are schematically shown in Figure 11. A model was selected to calculate the pressure in groundwater, and the pressure in the starting fracture was entered. Then, pressures were obtained at the point where the cracks emerged on the surface.

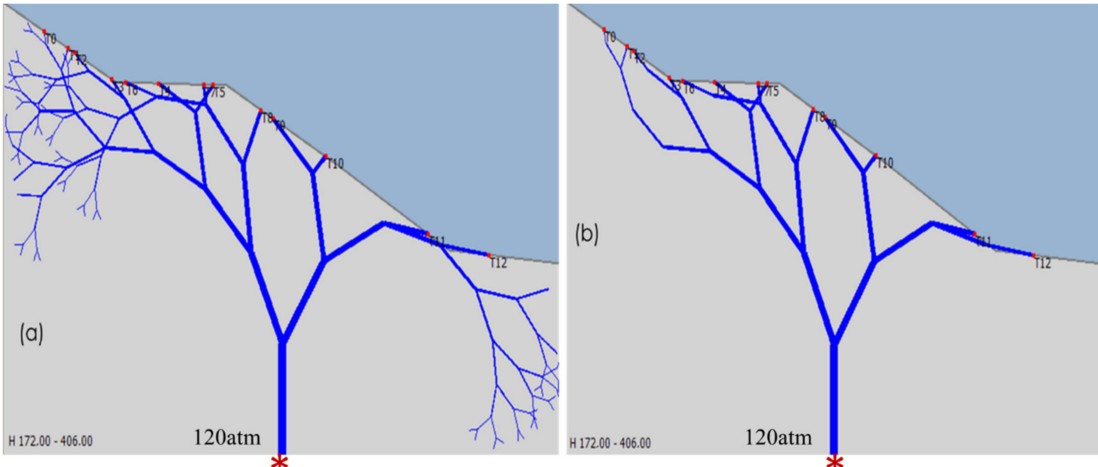

**Figure 11.** Computer simulation program for modeling groundwater pressure maps in fractured rocks: (**a**) dilution procedure; cracks that do not go out are excluded; (**b**) all cracks come to the surface.

The parameters set the coefficients for constructing the fractal tree. The numbers show the pressure quantities on the land surface when the initial source marked by a red star shows 120 atm as an induced pressure in the underground horizon. Pressure distribution data (marked by letter T with a digit), but only for the 7 outlet cracks that came to the surface, are presented below:

Initial pressure: 102000000 Pa.
T0: 2255009,15121366 Pa
T1: 2403699,590101134 Pa
T2: 2318031,00487013 Pa
T3: 2547345,0097631 Pa
T4: 5482358,92737351 Pa
T5: 5496730,73606538 Pa
T6: 5559331,83642009 Pa
T7: 5579804,96799125 Pa

Thus, a model was selected to calculate the pressure in the groundwater basin using the pressure in the starting fracture being entered—in practice, it should be measured by some instrumentation. Then, pressures were obtained at the points where the cracks emerged on the surface (vs. the 3D crack topology): ~dozens atm on the land surface (Figure 11).

This is a huge value. In fact, only about ten atms is enough for the destruction of artificial coating in concrete and asphalt on coated reinforced roads, caused by breakthroughs from underground waterpipes (cf. [14,22]).

We can recall a similar natural event, e.g., a geological phenomenon that was witnessed in the Indian state of Haryana (21 July 2021) in the north of the country, when the flooded land suddenly began to rise above the lake-water level. It became covered with cracks and swells (dynamic video is presented in [31]) caused by the sudden change in groundwater pressure. The situation is typical for mud-volcano eruptions [1,2,17,32].

All these processes can be analyzed in the simple hydrological models of the pneumo-hydraulic system (cf. [2,5,13,22]).

## 3. Results

As to the results of the considered approach first, we discuss a short statistical analysis "by measurements" of several parameters for some localized 3D river basin areas using groundwater state indication, with a proposal for how frequently it is necessary for the parameters to be measured in time: river discharge, precipitation level, and artesian water level in wells. First, we talk about key parameters in the database concerning these problems, obtained with different measurement procedures, without which, it is impossible to carry out any statistical analysis.

Second, within the confines of the general approach, we account for the basic consequences of possible tectonic influence on groundwater-transportation-system functionality.

Resulting from the reconstruction of the 3D crack-net under external factors, we also briefly discuss the practical verification of the risks of catastrophic flood development, applying the proposed approach.

All necessary parameters used for our consideration were obtained through official database analysis (see [7,10,11,25,26,33–35]).

### 3.1. Statistical Analysis

The key items for this study are:

- Independence/coherence/steady state of each process development according to its internal laws, as determined by autocorrelation function;
- The processes of correlation and mutual interaction being demonstrated in pair/crossed combinations;
- The same correlations but with different time shifts due to obvious and reasonable delays between different processes by selecting optimal time-shift as an adjustable parameter;
- Forecasting procedure with predictable parameters in time for the studied processes based on known/measured initial/fixed values.

We have carried out such procedures within the frame of the basic approach via general numerical statistical analysis concerning over 30 water events, but only 3 catastrophic floods in the USA river basin are presented as examples (Mississippi/Missouri (2011, Louisiana State), Boulder Creek (2013, Boulder County, Colorado State) and Santee (2015, South Carolina State)) because of the available/necessary data that were used for them (cf. [11,13,19,21]).

The subjects under consideration:

(i) Discharge and precipitation—are under season variations;
(ii) Groundwater—relatively speaking, is not directly correlated to season specifies;
(iii) Correlations/anticorrelations—do exist for such parameters as discharge mass, groundwater state and precipitation level.

As a result, we have made local conclusions over the data analysis in both different areas and time intervals for the observed water events in the form of obtained correlations. The objects and procedures of statistical correlation analysis are well known from textbooks; therefore, only the obtained final results for the four specific events are presented in Figures. displayed in Figures 12–14 and in Table 2 (database used from [7,10,11,25]). They are as follows:

(1) During catastrophic floods: the peak correlation of both precipitation (the Mississippi/Missouri region (no flooding simultaneously)) and discharge (on July 2011) were observed, but, as for groundwater level, the process of downfall occurs only in a single month (August 2011), and it has not recovered even in 2 years.

(2) As to autocorrelations for each unit: Strong for groundwater but weak for both precipitation and discharge take place.

(3) For mutual/pair correlations in a more detailed analysis we received:

- Negative correlation/anticorrelation coupling for groundwater and discharge in general, but it did not couple directly during the flood;
- Positive correlation coupling for precipitation and discharge but with some variations in time;
- No direct correlation coupling for groundwater and precipitation at the same time interval.

(4) We recognized a pair correlation of the processes with a temporal shift (±over several months) and did optimization by searching for the maximal correlation for the river basins: 1 month for the Mississippi and the Missouri, but 3 months for Santee.

(5) Regressive multifactor analysis was carried out with 0.33% accuracy for local data in comparison with averaging all data.

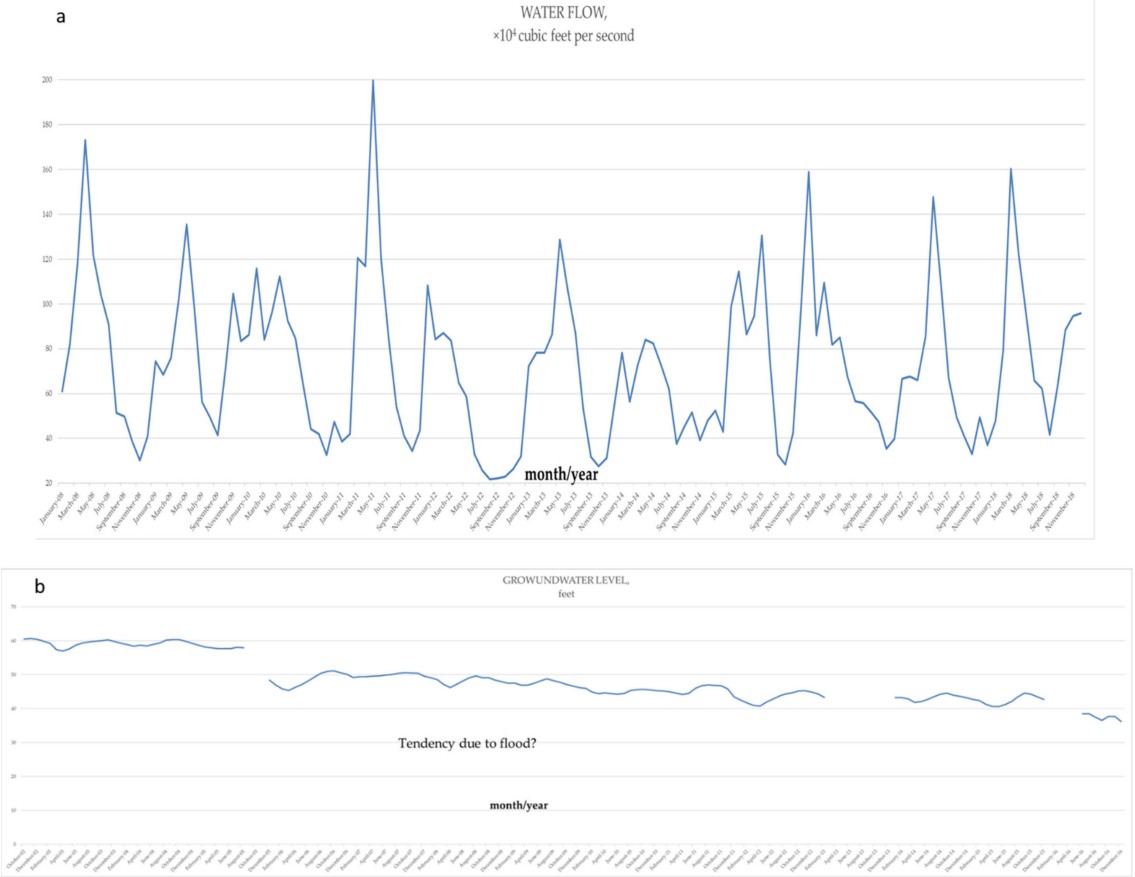

**Figure 12.** *Cont.*

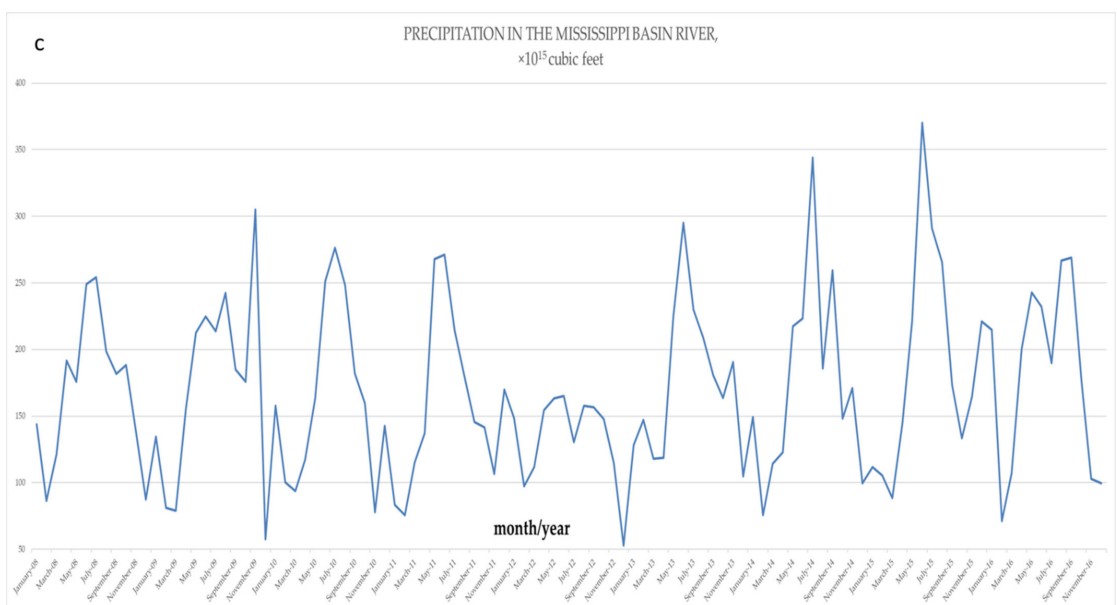

**Figure 12.** The Mississippi River, near New Orleans, Louisiana. (**a**) Monthly statistics graph based on water flow rate data in the Mississippi River. Water discharge behavior in the Mississippi River during the period from 1 January 2008 to 31 December 2016. (**b**) Monthly statistics graph based on the water table data of the Mississippi River. The groundwater level behavior in the Mississippi River from 1 January 2008 to 31 December 2016 (positive correlations). (**c**) Monthly statistics graph based on Mississippi basin rainfall data. Precipitation amount behavior in the Mississippi basin from 1 January 2008 to 31 December 2016. We received positive pair correlations for (**a**,**c**) and negative correlations for (**a**,**b**); the facts probably demonstrate a tendency to flooding. On the horizontal axis–the breakdown of data by year/month. On vertical axes–(**a**) water discharge (in feet/sec); (**b**) groundwater level (in feet); (**c**) precipitation level (in cubic feet).

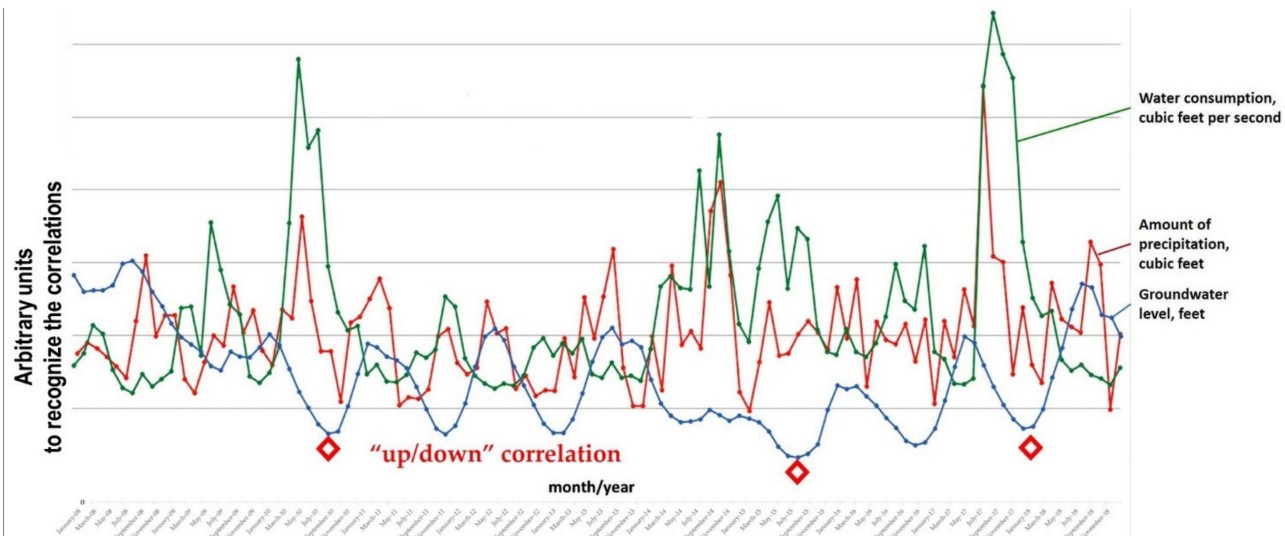

**Figure 13.** Statistics for the Santee River, South Carolina ("up/down" correlation for water consumption and groundwater showing the local reduction of underground reserves due to flooding). On the horizontal axis–the data by year/month. On vertical axes–volumes of water for precipitation (in cubic feet), water consumption (in cubic feet per second) and groundwater level (in feet). Red diamonds mean a noticeable anticorrelation of groundwater level with water masses in the form of precipitation and water consumption.

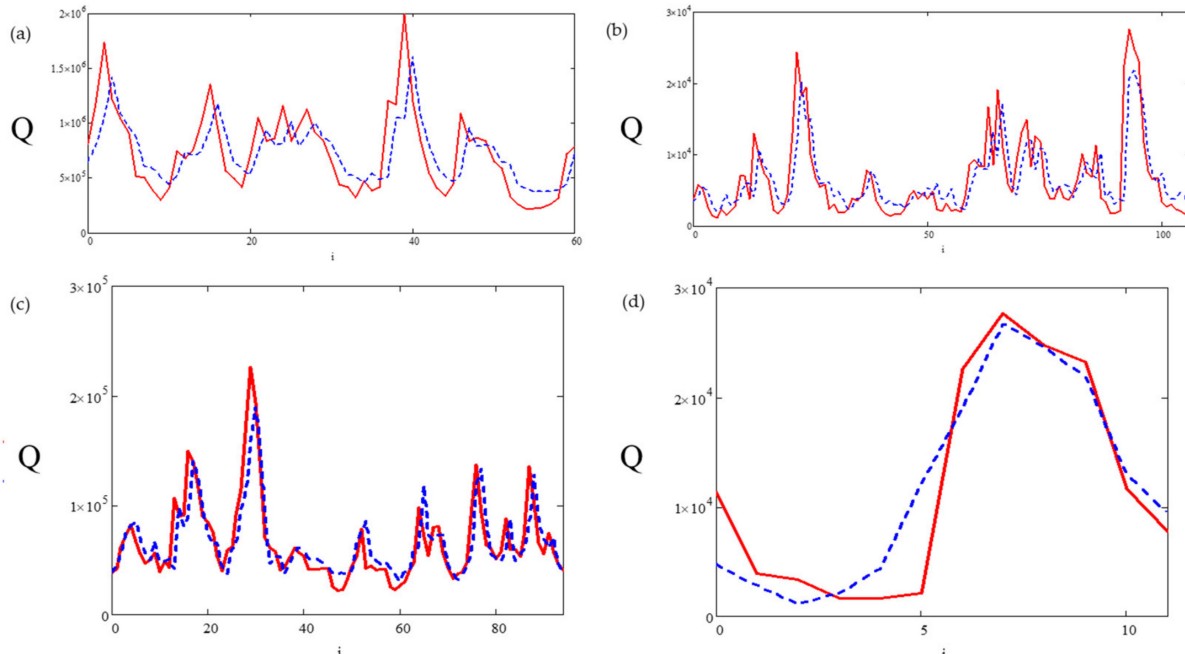

**Figure 14.** The results of mathematical modeling for flood forecasting based on statistical data for the entire research period (i–number of months): (**a**) for the Mississippi river; (**b**) for the Santee river; (**c**) for the Missouri river; (**d**) more detailed scale fragment for the Santee river (see text above the figures). Here, Q–real water flow (feet/s) (red) and predicted dependence (blue).

**Table 2.** Calculation results of the visibility coefficient γ (showing the process of correlation in time for the units) based on the data for water consumption/discharge, ground water level, and the amount of precipitation in the study areas.

| River | γ for Water Flow (Discharge) | γ for the Ground Water Level | γ for Precipitation |
|---|---|---|---|
| Mississippi (May 2011) | 0.89118678 | 0.271321887 | 0.857674013 |
| Boulder Creek (September 2013) | 0.996339325 | 0.220981998 | - |
| Santee (October 2015) | 0.959963899 | 0.547425876 | 0.900993342 |
| Missouri (2011) | 0.901901813 | 0.653395031 | 0.857674013 |

The corresponding graphs are shown in Figures 12–14.

A graphical analysis for the Mississippi River shows that, in the 9-year observation interval within database, for each year, with the exception of 2012, its activity correlation peaks were tracked. In the case of flooding in 2011, these correlation peaks occurred in May, being the month of flooding (Figure 12).

The results of data processing for the Santee River are presented in Figure 13.

The water consumption forecast at future points in time (*t*) was carried out based on dependence on the current database—water consumption, groundwater level and precipitation intensity—by the formula:

$$Q(t + \Delta t) = f(Q(t), h(t), P(t))$$

where $Q(t + \Delta t)$–forecast of water flow through $\Delta t$ time periods; $Q(t)$: current consumption; $h(t)$: the current groundwater level; $P(t)$: the amount of precipitation at the current moment.

We carried out an adjustable procedure for fitting the correlations of different types for the 3 mentioned above rivers (Figure 14). On the vertical axis—the solid red line $QS_i$, marks the statistical by real maximal flow of the rivers in the flood years. On the vertical axis—the italic blue line $Q_{t+\Delta t}$, marks the predicted maximal water flow rate. As to the i-index, it indicates the number of months for the analysis made.

The generalized correlation data for the time-dependent oscillations of the processes are presented in Table 2. The analysis is carried out by the visibility $\gamma$-coefficient for maximal ($I_{max}$) and minimal ($I_{min}$) water level: $\gamma = (I_{max} - I_{min})/(I_{max} + I_{min}) = [0,1]$. In the dynamic oscillatory process, we received $\gamma \to 0$ for dip variations and/or $\gamma \to 1$ for stable/steady-state (~constant level).

Finally, let us briefly discuss the analysis procedure, i.e., for the Mississippi river, 2011.

First, let us compare two factors: discharge and the precipitation level from 1 January 2011 to 31 December 2011. For correlation coefficient K ("day by day") we received an unexpectedly very small value of K $\sim$ 0.011 (maximal discharge period was during May and exceeded the usual level, e.g., in February, 7 times).

Second, as to correlation coefficient K between precipitation and groundwater level, its value was small as well (K $\approx$ 0.060), but the groundwater level did not sufficiently vary during the whole of 2011 in contrast to the precipitation intensity for the same area of approximation. This means that precipitation does not directly impact immediately (we forget here about different the localization of stations in the areas under measurement). To adjust the day shift parameter for the maximal value of the correlation coefficient, we increased it and attained the values for two discussed cases, though not more than K $\sim$ 0.7.

The correlation between discharge and groundwater levels during the flood, with several days' shift, is shown in Figure 15. The results are the following.

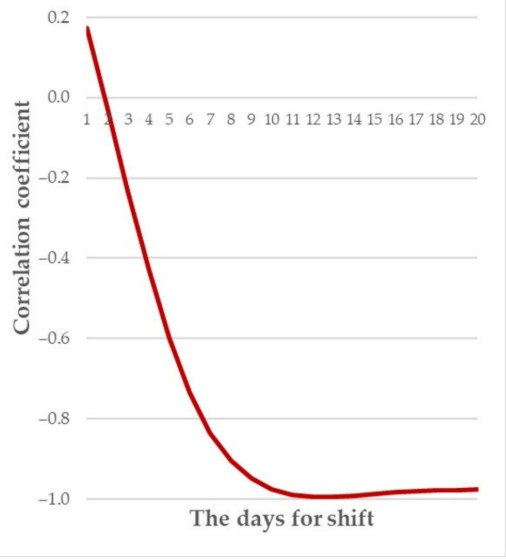

**Figure 15.** Correlations (vertical axis) between river discharge and groundwater level vs. selected days' shift (horizontal axis).

In quiet seasons (before the flood) the correlation coefficient K between these two factors (discharge and groundwater level) is K = −0.74 (anticorrelation events), which means that an increase/decrease in river discharge depends on a decrease/increase in the artesian water level. These natural cycles in time are typical for a river basin area in an equilibrium state.

When the flood occurred (May 2011) we had K $\sim$ −0.50 for the measurements made "day by day".

However, with the day shift over 13 days (pre-event days were fixed for different events), we received practically absolute correlation: $K_{13} \sim$ −0.994 for a distance of ~200

km (according to station sourced for database collection), i.e., artesian water obviously resulted in surface-water discharge increase (see Figure 15).

However, all these conclusions are relatively problematic and show trends because, first, they strongly depend on the averaging scale for available data. The procedure of the averaging scale for available data means that these data were taken at fixed points in time and localized areas for different spatial locations where the monitoring stations are located. Second, the discharge parameter was determined not only by water mass itself but flow velocity in general. Third, the correlations between different processes strongly depend on the temporal shift in days (both natural and modeling) for the events—observable and calculated. Fourth, the dislocation of the stations, being the resources of the database, cannot be controlled absolutely in the same studied areas.

### 3.2. Earthquake Impact

Systematized results in possible schemes of earthquake impact by the wave propagation process are shown in Figure 16 (according to database [26,33–35], cf. [13,21,27]).

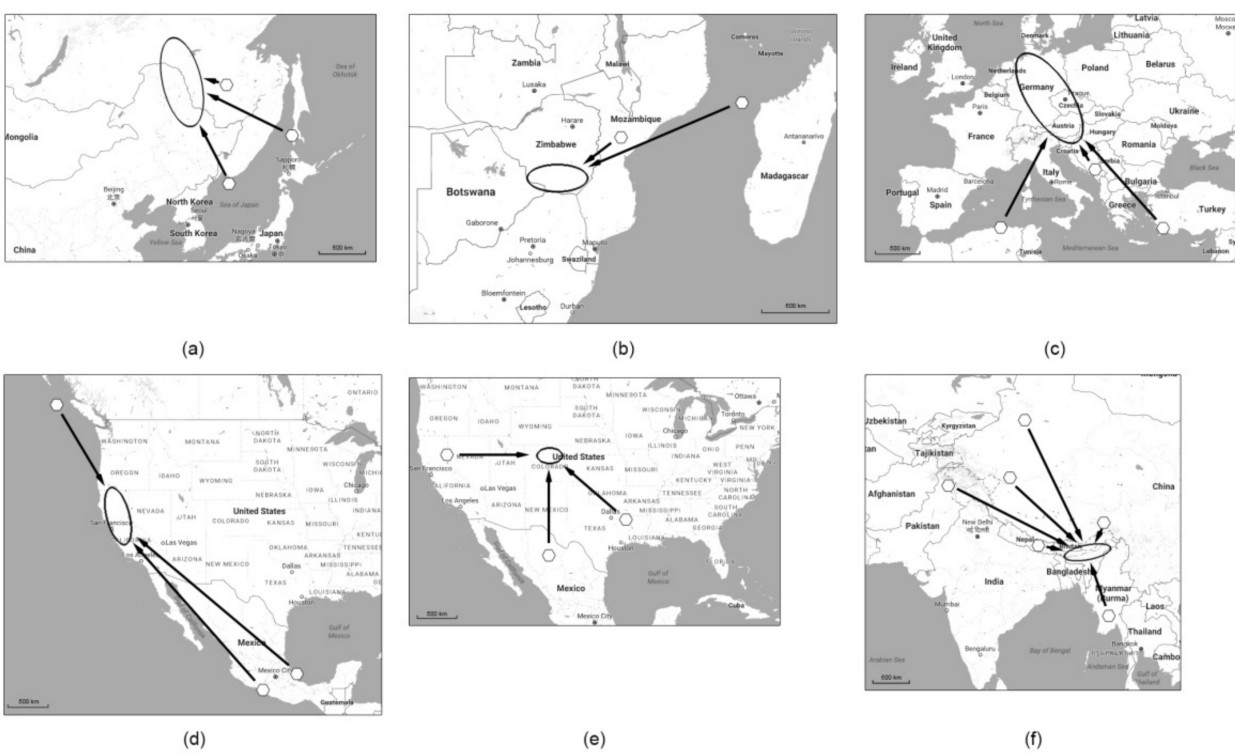

**Figure 16.** Relative positions of groups of earthquake epicenters regarding the flooding area: (**a**,**b**)—one-directional arrangement; (**c**,**d**)—two-directional arrangement; (**e**,**f**)—multi-directional arrangement. White hexagons—the earthquakes epicenters; black ovals–the flooding areas.

A special case is the 2013 (12–15 September) Boulder Creek, Colorado, catastrophic flood (Boulder County) [18,25]. In fact, in this case, long-lasting heavy rains (430 mm of precipitation) resulted in a water discharge increase in Boulder Creek, from 5 m³/s to 140 m³/s, which was unexpected due to both the great value and large area of the water accumulation for localization, that is, in such a small riverbed without taking into account coupling with the flash process of the groundwater exit. In addition, if we take into account preceding earthquakes (4.2 magnitude, North California (27 August 2013); 4.3 magnitude, North Mexico (28 August 2013) and 4.5 magnitude, East Texas (2 September 2013)), then the event becomes understandable due to a reconfiguration of the crack-net for groundwater exit.

Previous research concerning the interconnection of floods and preceding earthquakes has an even brighter example of such a manifestation for a similar case because of the constructive interference of three different seismic waves which were probably focused on one point of location (for simplicity, we have presented the circular seismic wave fronts)—see Figure 17 and the associated Table in the right upper corner. If the hypothesis of interconnection between floods and preceding earthquakes is true, the 2013 Colorado flood was obviously predictable [18,36].

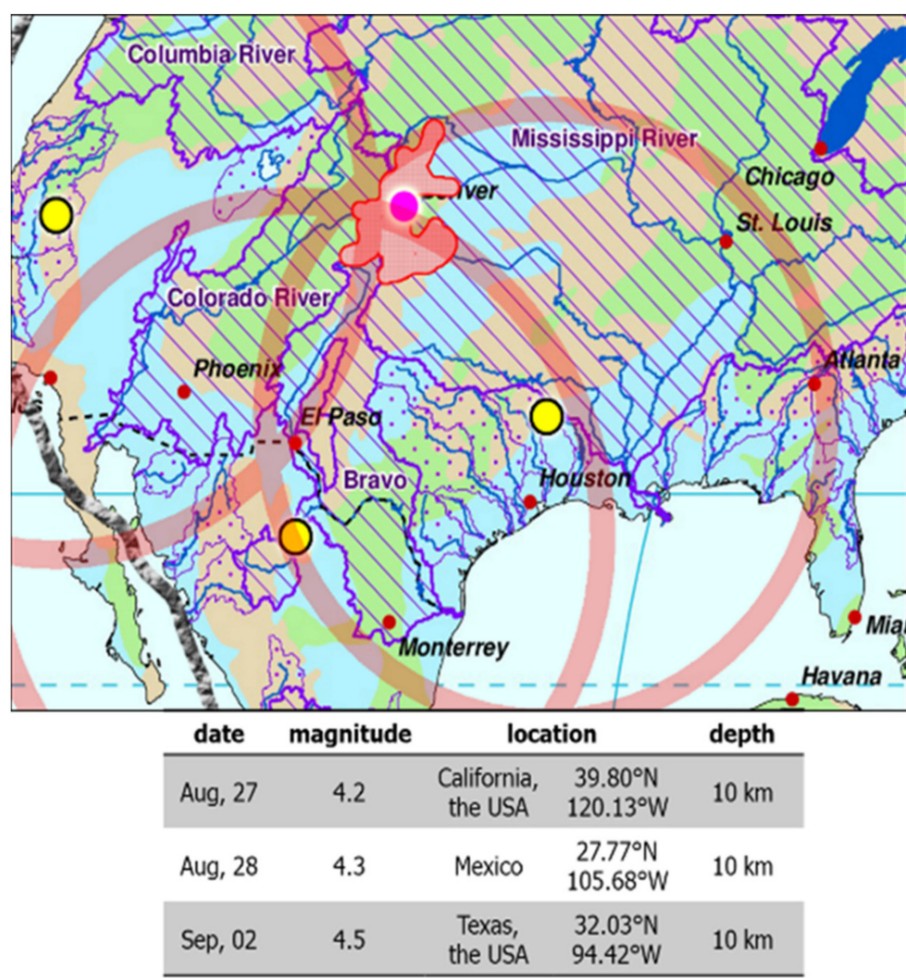

| date | magnitude | location | | depth |
|---|---|---|---|---|
| Aug, 27 | 4.2 | California, the USA | 39.80°N 120.13°W | 10 km |
| Aug, 28 | 4.3 | Mexico | 27.77°N 105.68°W | 10 km |
| Sep, 02 | 4.5 | Texas, the USA | 32.03°N 94.42°W | 10 km |

**Figure 17.** The Boulder County (Colorado, USA, purple circle on the map) event, located exactly where three wave circles cross, with centers in the earthquakes' epicenters (yellow circles on the map); i.e., this region has experienced a great conflict of seismic waves.

## 4. Discussion

A significant increase in runoff volume causes the depletion of groundwater resources at the end of a flood when this resource has been depleted for some period. The duration of this period is defined by the groundwater recharge rate in specific geological, geographical and climatic conditions.

One more important aspect of such depletion of groundwater resources is connected to the strange factor of increased wildfire risk in the future. In fact, for example, the flooding in California, USA, in February–June 2017 lasted for half a year, and afterwards, large wildfires occupied the state and lasted for two following months (see Figure 18) [36]. This is possibly connected to insufficient soil moisturizing after the flood as water goes to balance the recovery of deeper aquifers.

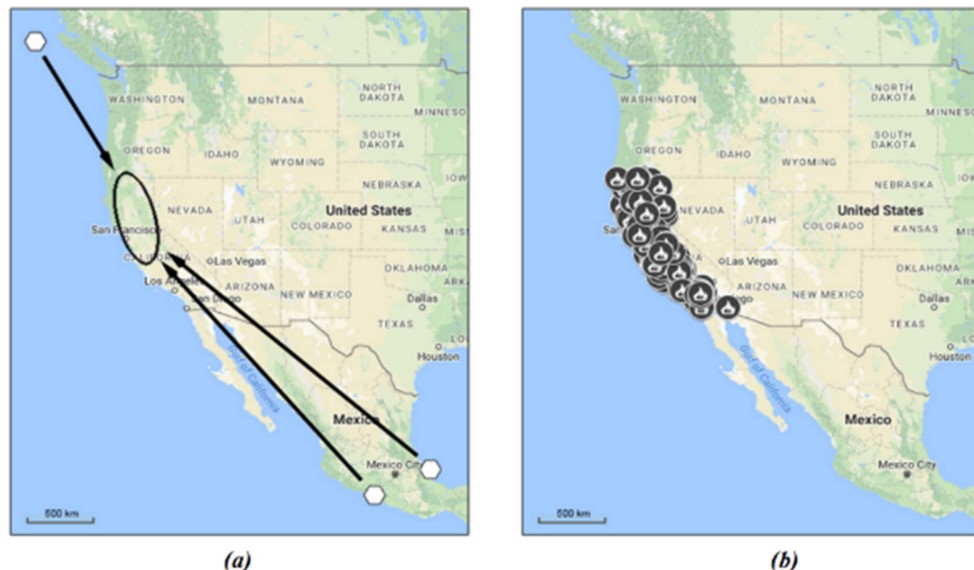

**Figure 18.** Unexpected consequences of disastrous floods in the USA (2017). (**a**)—white hexagons–the earthquakes epicenters; black oval–the flooding area; (**b**)—the wildfires seats.

Another feature concerns the hydrostatic pressures map in the 3D crack-net of the river basins, similar to the system for communicating vessels. In fact, for example, when the flood in the Amur River basin (2013, Russia/China) is analyzed [14], the neighboring surface river basins of the Amur and Lena (Russia) rivers can be considered to be connected because of a possible common source in an underground basin. Moreover, simultaneous to the disastrous Amur flood, the phenomenon of water level falloff in the Lena River below the navigable level was observed [37].

Indeed, our analysis provided similar results concerning floods in Kazakhstan and the Tyumen region (Russia) in spring 2017 [38]. The same phenomenon occurred in the surface basins of the Ob River (Russia), where the flood developed, and the Yenisei River (Russia), which abut to each other. Thus, large wildfires along the Yenisei River basin are more likely to occur because of the simultaneously development of a flood on the Ob River. It is, evidently, natural for us to take into account the depletion of the common groundwater basin of these rivers (see Figure 19).

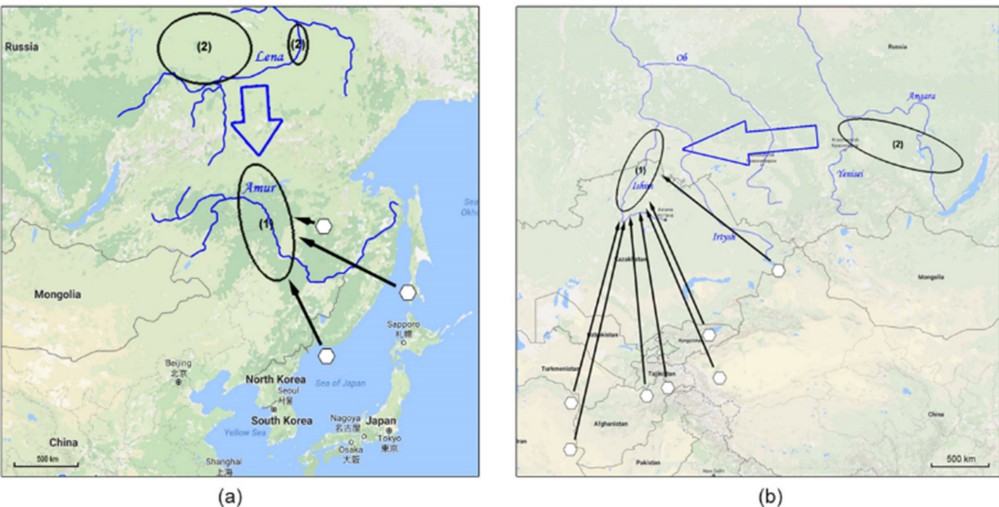

**Figure 19.** The river basins' interconnection. (**a**)—the Amur and the Lena Rivers, (**b**)—the Ob and the Yenisei Rivers. White hexagons—the earthquakes epicenters; black ovals (1)—the flooding areas; black ovals (2)—the areas of wildfires propagation.

This is why connections between underground basins of different (great) rivers may be a global phenomenon on a geological scale [39], but the process is dynamic, and earthquakes may play a universal role in coupling phenomena over great distances (see Figure 20).

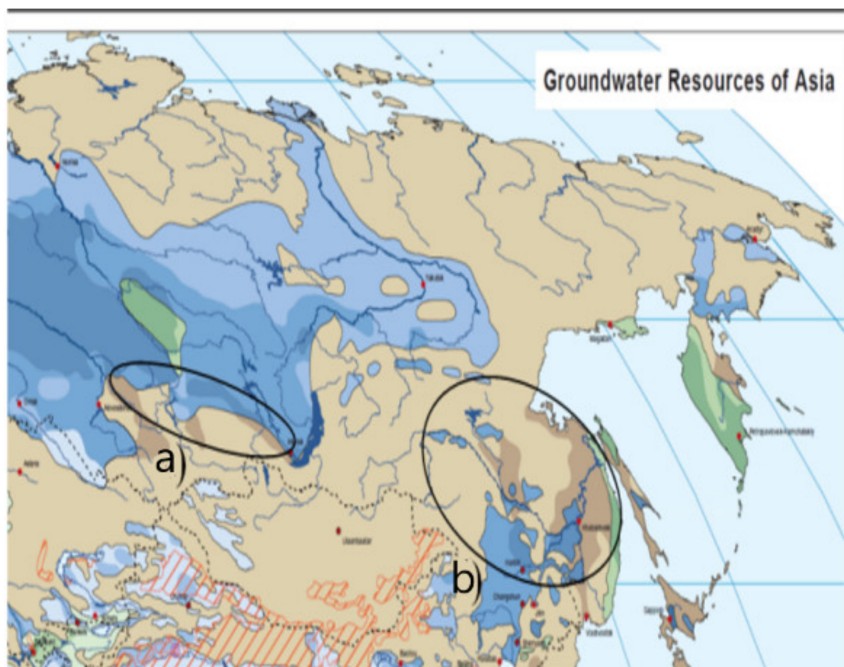

**Figure 20.** A groundwater map of both the Amur River channel and the upper reaches of the Yenisei River (marked by closed areas), which lie close to the earth's surface and are characterized by instability in the hydrological regime: (a) at the junction of the Baikal and Caledonian folding; (b) at the junction of Baikal, Herzian and Mesozoic folding (according to the World-Wide Hydrogeological Mapping and Assessment Program).

Detailed and possibly quantitative analysis is a matter of future study.

In addition, we carried out a computer simulation of the trigger water discharge/mudflow process from underground up to the surface using the soliton nonlinear hydro-dynamic model (see, e.g., [24]), being a multi-developed structure in dynamics, caused by propagation along the inclined surface (cf. [13]). The process from the very beginning was under the thixotropic effect, reducing the liquid mixture viscosity under vibration for various reasons (cf. [2,22]), e.g., due to microseismic effects—Figure 21. This is a natural dynamic consequence for the debris event occurring due to the sudden reconstruction of the 3D crack-net near the surface, and local flash discharge of groundwater.

Our analysis shows that the mudflow process can be represented by four-stage development and propagation for a mudflow soliton: (1) the main mudflow discharge occurs there; (2) the process falls into separate soliton satellites; (3) it is the stage of self-organization for these satellites according to the values of their amplitudes in the propagation process; (4) the soliton is breaking, i.e., turning over (great nonlinearity) or a decay (great dissipation) process takes place.

The model is probably applicable to the Crymsk City debris event (Russia, the Caucaseus region), 2012, (cf. [8,13]), and may be reasonably applied to any debris event in a mountainous landscape under stochastic processes of a different nature [40,41].

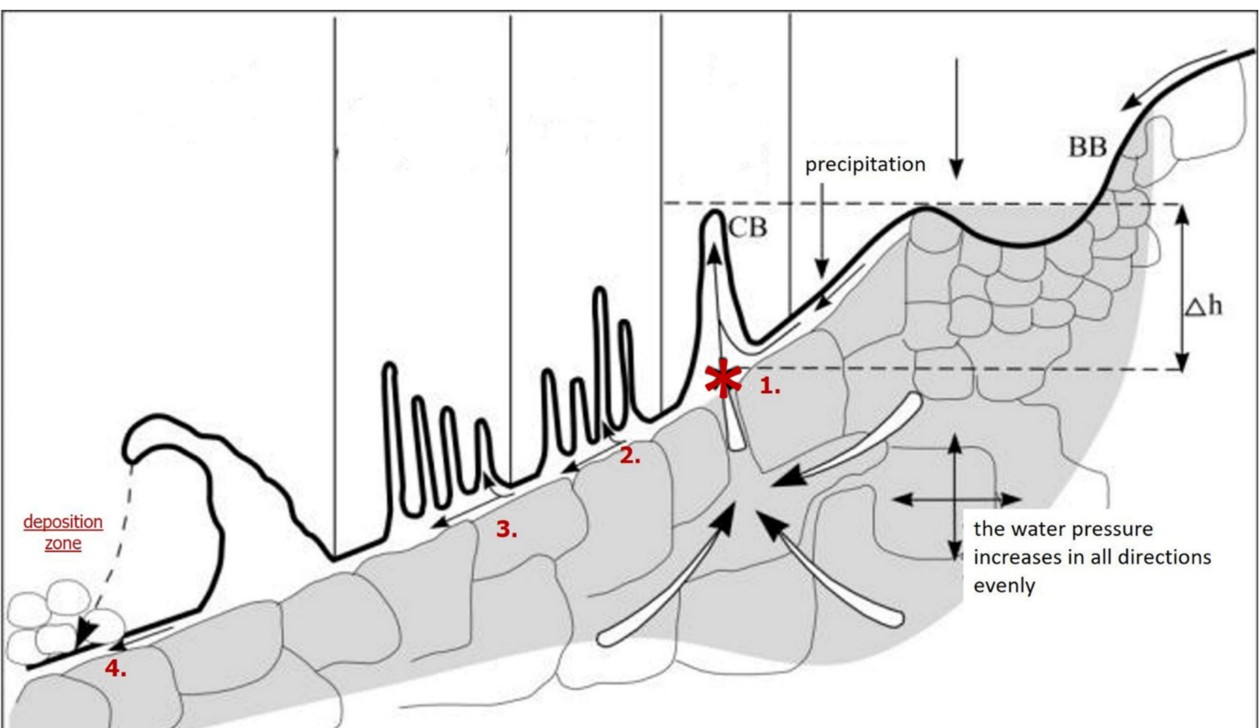

**Figure 21.** Computer simulation of trigger water discharge/mudflow process—soliton nonlinear hydrodynamic model. The multiple solitonic variation regime propagation is shown and developed from a single soliton from the very beginning due to pressure variation over the induced stable channels in the void cavity system (the discharge/mudflow exit on the surface is indicated by the red star, 1). BB–collecting funnel; CB–mudflow soliton wave; $\Delta h$–hydrostatic thrust/pressure head; 1.–mudflow gate; 2.–surface water with drainage process contribution; 3.–multisoluton movement; 4.–final surface flows.

## 5. Conclusions

According to our study, we summarize the obtained results in the following aspects as a practical verification for the proposed approach based on an objective database of the events (e.g., presented in [7,10,11,25,26,33–35,42–45]).

1. Based on the discussed model, forecasts with vital information about both ground-water hydrostatic/hydrodynamic pressure distribution and water flows, carried out by a water crack 3D map in mountain massifs, should be introduced into theory and analysis.

2. A necessary condition for the dramatic development of the phenomena is the breaking down of impermeable rock caused by sudden openings in crack-ways (previously blocked), that become active for some reason; e.g., due to shower runoff impact, geo-thermal stream influence, or earthquakes.

3. The water from the top hill-lake/reservoir and/or down-lake/reservoir (local base level) can reach the below and/or upper river area (the base level) via the activated groundwater transportation routes due to connecting vessels affected by the development of a backwater process because of intrinsic pressure variation.

4. Traditional and artesian wells, being preliminary and artificially made by a certain topology strategy, bring up an opportunity to formulate water cracks with hydro-static/hydrodynamic pressures in the 3D map of the mountain massif; i.e., a recognition of the water flow physical state for modeling. This approach results in knowledge of real parameters for modeling and, finally, for a forecast map design taking into account the necessary databases by satisfying the greatest challenges for acceptable risk estimation and early warning systems.

Additionally, within the framework of this concept, graphic illustrations of floods in Europe (2013) are shown in Figure 22 (see also Figure A1 in Appendix A below), and the corresponding explanations are presented. Thus, based on our proposed approach, it is possible to assess potentially dangerous areas for preliminary predictions of possible catastrophic floods.

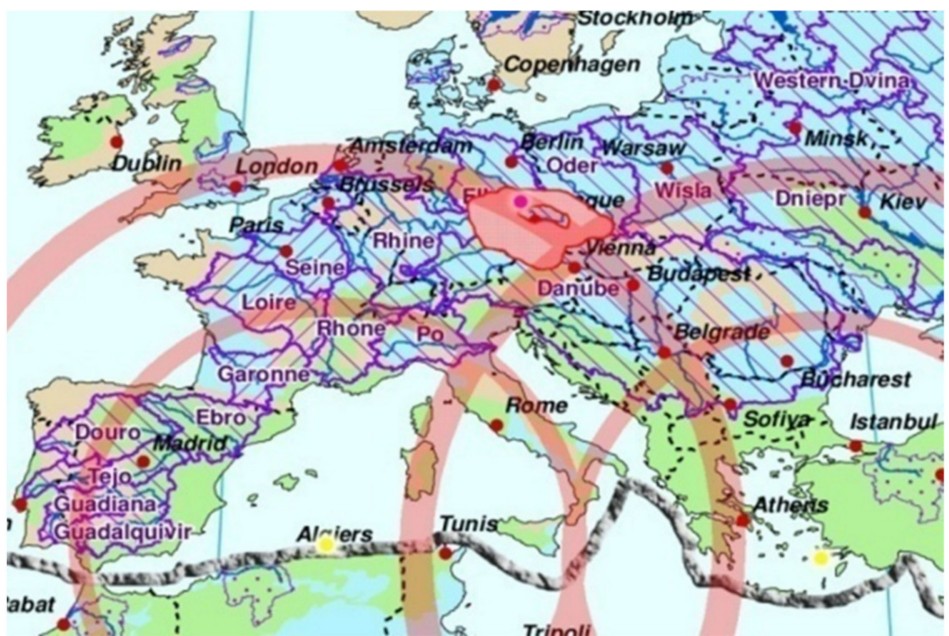

**Figure 22.** Flooding in Europe–05-06, 2013 (both without and with designation): yellow circles–earthquake epicenters; purple circles–fixed location of the flood areas; transparent red circles–schematic representation of seismic wave propagation; gray curves–lithospheric plate boundaries; areas with cranial border–potentially dangerous zones (marked by red color areas) for catastrophic floods based on seismic factor analysis in association with the river basin landscape.

As to a global reason for instability in 3D river basin states, this aspect may be explained by solar–terrestrial relations being a typical process with regard to the subject under consideration (see Figure 23 for statistical analysis of seasonal changes in M ≥ 7 frequency earthquakes from 1900–2004, presented as a percentage of the average value for 2659 events, by Prof. A. Yu. Reteum from Lomonosov Moscow State University—private report, according to, e.g., [33,35,46], and based on the data from the International Seismological Centre, 1990–2019, with an average of every 4 years). The discussion of the problem in different aspects is presented in [4,46]. However, practically, catastrophic water events often occurring simultaneously in different regions of the Earth supports this global thesis (cf. [7,10,11,26,33,35,45]).

Our analysis shows that the mudflow process can be represented by four-stage development and propagation for a mudflow soliton: (1) the main mudflow discharge occurs there; (2) the process falls into separate soliton satellites; (3) it is the stage of self-organization for these satellites according to the values of their amplitudes in the propagation process; (4) the soliton is breaking, i.e., turning over (great nonlinearity) or a decay (great dissipation) process takes place.

The model is probably applicable to the Crymsk City debris event (Russia, the Caucaseus region), 2012, (cf. [8,13]), and may be reasonably applied to any debris event in a mountainous landscape under stochastic processes of a different nature [40,41].

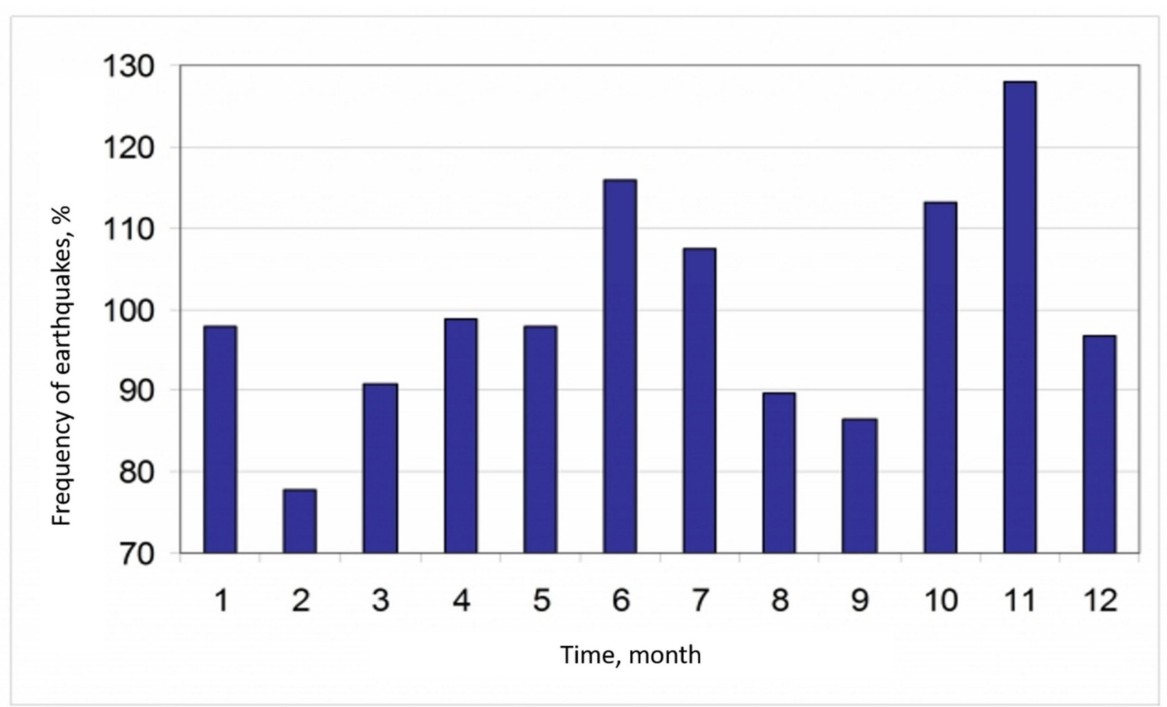

**Figure 23.** Analysis results for solar–terrestrial relations with regard to earthquakes occurring. Seasonal changes in earthquake frequencies of more than magnitude 7 for 1900–2004, in comparison to the average value percentage (%) over 2659 events for each 20-yr period. Months 6–7 and 10–11 of the year are usually the most dangerous for the occurrence of catastrophic floods.

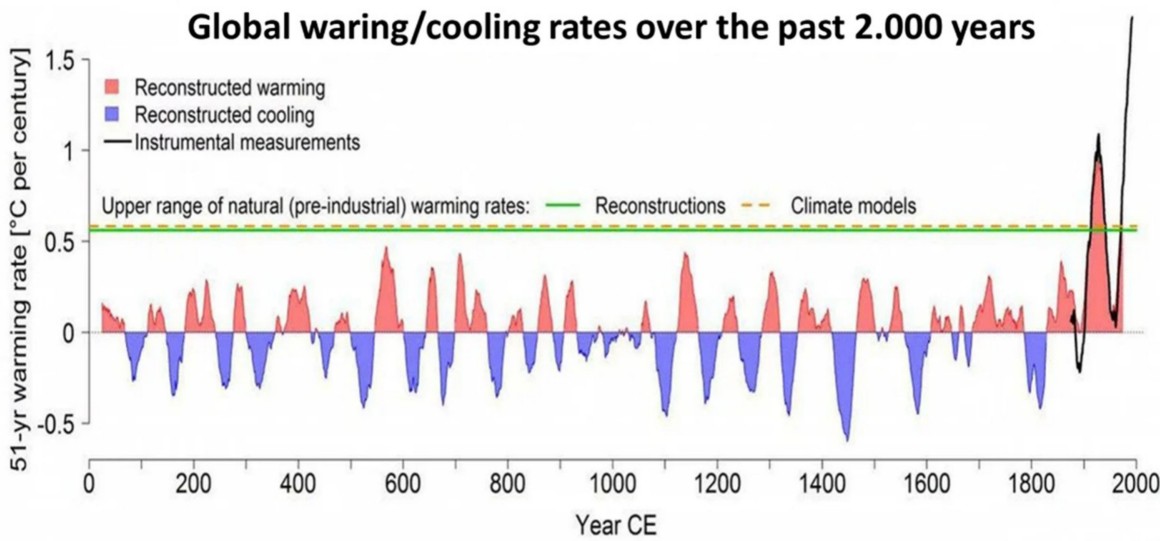

**Figure 24.** Natural time-scale dependence for the warming and cooling periods over the last 2000 years.

Thus, it seems to be reasonable to conclude that all global processes in the Earth lifecycle are determined by solar–terrestrial relations. However, such a fundamental approach requires more detailed study based on lots of reliable data and adequate modeling for different regions.

Finally, we note that, to consider the subject in progress, we have to overcome the problem of not having enough databases containing observable events to make adequate analysis, and thus, we need a better, new knowledge base concerning the development of events in different times and areas. Then, we can carry out simulation modeling within

the confines of methods for stochastic, nonlinear, dynamic processes by the manipulation of key uncertainty parameters involved in model (induced by many factors: precipitation, temperature, solar radiation, soil state, rock composition and structure, landscape relief and rivershed basin characteristics, crack-net structure, groundwater debit timing and mapping).

This will allow you to perform, firstly, a search for big fluctuation occurrences, resulting in the development a complex processes under the required conditions in a nonlinear stochastic wave system and also to study stability levels under external perturbations and/or the principal variations of vital parameters in such a system. Secondly, it will allow you to carry out predictive modeling using achievements in the quantum uncertainty physics approach and technologies for forecast procedures of complex processes based on many competive parameters.

In conclusion, we can formulate preliminary recommendations for the identification of earthquake influence on disastrous floods in a 3D river basin. Within the framework of presumably connected preceding earthquakes and historical/disastrous subsequent floods, the possible classification of probable conditions for these subjects may be grouped in the following ways.

Firstly, to make the analysis with regard to the relative positions of preceding earthquake epicenters:

(1) One-directional arrangement: it can be both a single earthquake and a group of earthquakes.

(2) Two-directional arrangement: the general case is the arrangement of epicenter groups at different distances and directions from the risk zone; in this case, an additional analysis of local geological structures and groundwater recharge rates is necessary (the example of a special case is the arrangement of epicenter groups at equal distances from the risk zone).

(3) Multi-directional arrangement from any earthquake source.

Secondly, to consider the factors influencing groundwater 3D-transport-net topology:

(1) Blockage of some parts with dramatic pressure rises in the net with a water-hammer manifestation on the surface.

(2) Connection/disconnection of groundwater basins (smoother development of flood; longer effect of flow).

Thirdly, to take into account the spatial scale of manifesting consequences:

(1) Local restructuring of 3D-transport-net topology that does not break the stable regime of river basin functionality.

(2) Significant restructuring of the 3D-transport-net topology that breaks the stable regime of the river basin's functionality and causing the water level to rise in the river, resulting in the flood.

(3) Significant restructuring of the river's 3D-transport-net topology that affects the common, unified groundwater basin, e.g., for two rivers, and causing a catastrophic rise in the water level in the river (for one surface river basin) and a fall in water level in the river (for another neighboring surface river basin).

These recommendations are preliminary but not exhaustive, as there are plenty of specific territory features that are outside of the considered classification, but which may play a key role in the emergence and development of disastrous floods. However, these recommendations are useful in the case of pre-forecasting probable disastrous water events as a recognition of the tendency and trends of their arising.

Finally, let us use the database for both volcanic activity and possible earthquake impacts on flood development as a preliminary hypothetical/speculated demonstration, summarized in Table 3 (for database, see [19,26,33–35].

**Table 3.** Tectonic processes, flood locations and probable coupling.

| Items | Selected Collection of Seismic Events/ and Data/and Magnitude | Proposed Related Flood/ and Data | Time Factor/ Time Delay for Coupling | Distance between Two Events (Coupling Scale) | Note |
|---|---|---|---|---|---|
| | I. Basic events/test events for establishment of the coupling | | | | |
| 1. | Nord Japan/ 26 April 2001/5.96 | Lensk (Yakutiya, Russia)/ 14 May 2001 | 18 days | $2.2 \cdot 10^3$ km | (1) artesian cracknet with spatial distance of groundwater coupling—about few thousand km (2) sudden modification of the 3D-crack topology and resistance against the fluid flows |
| 2. | Nord Taiwan/ 14 June 2001/5.87 | Kultuki (Irkutsk region, Russia)/7 July 2001 | 23 days | $3.4 \cdot 10^3$ km | |
| 3 | Afghanistan/ 3 January 2002/6.05 | Temruke (Krasnodar region, Russia)/ 10 January 2002 | 7 days | $2.9 \cdot 10^3$ km | |
| | II. Verification of the proposed coupling (events at present) | | | | |
| 4. | Popocatepetl Volcanic eruption (Mexico)/ 5 July 2013 | Ruyaya State (Mexico)/20 July 2013 | 15 days | $1.3 \cdot 10^3$ km | |
| 5. | (a) Instability Land Cluster in time: Sakuradzima Volcanic eruption (Kyushu island, Japan)/ 10 July 2013/ emission of ash from the volcano up to 3 km height; (b) Izu Archipelago (Japan)/ 11 July 2013/ 5.3; (c) Nord-East Honshu island (Japan)/ 13 July 2013/ 4.5 | Nord Honshu (Japan)/ 18 July 2013 | 5–8 days | $1.9 \cdot 10^3$ km $0.9 \cdot 10^3$ km $0.2 \cdot 10^3$ km | Should be the flashy flow process due to the ground pressure sudden enhancement ~1000 atm |
| 6. | Kamchatka (Russia)/ 17–18 July 2013; Volcanic Shiveluch/ on July 2013 | Ivanovka (Amur region, Russia)/ 20 July 2013 Kamchatka (Russia)/ 29 July 2013 | 3 days 1–3 days from last eruption | $5.5 \cdot 10^3$ km $0.4 \cdot 10^3$ km | Continuous Earth-quake vibrations result in 3D-reconstruction of crack-net in continuous dynamics |
| | III. Neural-Net training | | | | |
| | In progress | | | | Needs a reasonable database |
| | V. Forecast for acceptable risk | | | | |
| | In progress | | | | Final goal: The risk mapping design in both space and time |

These kinds of selected events, under the analysis of the possible coupling in tectonic processes and floods, may be presented as an adjustable preliminary catalogue for future study. In fact, our analysis shows that the strategy for such correlation between the localization of earthquake/volcanic eruption and ongoing floods on river basins may be affected in the latitude of 30–50°, with no more than 180 days delay in time and in a fixed, minimally estimated distance between these two phenomena in space.

**Author Contributions:** Conceptualization, S.A. (Sergei Arakelian) and T.T.; methodology, M.A., D.B. and S.A. (Sergei Abrakhin); software, D.B. and S.A. (Sergei Abrakhin); validation, S.A. (Sergei Arakelian) and S.A. (Sergei Abrakhin); formal analysis, T.T. and S.A. (Sergei Abrakhin); investigation, S.A. (Sergei Arakelian), M.A., S.A. (Sergei Abrakhin); resources, M.A., D.B. and S.A. (Sergei Arakelian); data curation, D.B. and S.A. (Sergei Abrakhin); writing—original draft preparation, S.A. (Sergei Arakelian) and T.T.; writing—review and editing, S.A. (Sergei Arakelian) and S.A. (Sergei Abrakhin) and T.T.; visualization, S.A. (Svetlana Abrakhina) and D.B.; supervision, T.T.; project administration, S.A. (Sergei Arakelian); funding acquisition, S.A. (Sergei Arakelian). All authors have read and agreed to the published version of the manuscript.

**Funding:** This research received no external funding.

**Informed Consent Statement:** Informed consent was obtained from all subjects involved in the study.

**Conflicts of Interest:** The authors declare no conflict of interest.

## Appendix A

Now we present several objective databases and illustrations helpful to understanding the basic concepts of our approach. The principal background database platform can be found in [40,41,49–56], and is useful for initial analyses.

The demonstration according to our model:

1.  In the enclosed figures, we show the water reach/risk area for accidents with regard to groundwater and surface lakes interacting with pressure variation in the 3D crack-net of a river basin caused by both heavy rains and seismic activity [10–12,25,26] (see also [35,42]).

2.  If we talk about liquid/groundwater movement in cracks with a small cross-section, the speed of such movement strongly depends on fractured rock composition, which leads to a paradoxical result where a more viscous mixture has higher velocity (see [53–55]). This issue with hydrodynamics and related phenomena (see [1–3,17,34,56]) requires separate consideration for each specific underlying surface case, in association with the discharge and debris processes.

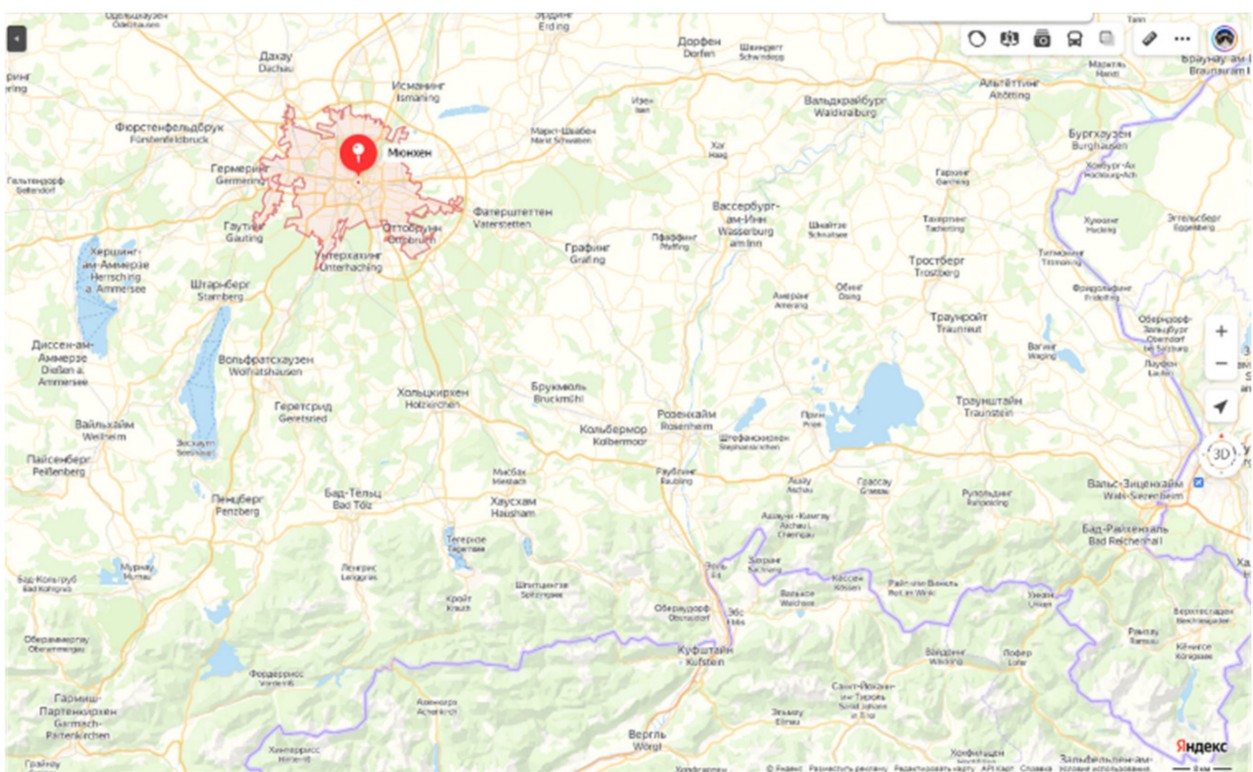

**Figure A1.** Due to heavy rains since 12 July 2021, the tributaries of the Rhine Ar and Moselle, as well as several smaller rivers, have overflowed their banks in the west and southwest of Germany. The main impact of these elements fell on the lands of North Rhine-Westphalia and Rhineland-Palatinate.

Finally, it is interesting to note that groundwater-state monitoring is possible by reaching depths of 10–20 km using novel drilling technology [57].

Some practical approaches to optimize control policies for reducing urban drainage flow generated by some methods outperform in both peak flow reduction and rainwater availability, as considered in [58].

This approach (and the related concept) may result in more accurate forecasting and early warning systems for catastrophic water events in the form of Emercom Agency activity (cf. [59,60]).

Finally, to study both the dynamics of the groundwater lifecycle and the natural background processes of water horizons, it is reasonable to use a database and different

protocols and approaches for the measurement of the movement of potentially toxic compounds as a possible instrument of monitoring a water way's distribution in a system in order to make a forecast.

These methods are very well developed for the subject of groundwater management in a practical sense (see, e.g., [61]).

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
