# Peer review of "Catastrophic Floods in Large River Basins: Surface Water and Groundwater Interaction under Dynamic Complex Natural Processes–Forecasting and Presentation of Flood Consequences"

_water, doi:10.3390/w14091405_

Round 1
Reviewer 1 Report
It seems that the authors did not attached their final version of the manuscript - as these yellow fields significantly degrade the readability of the text.
However, the text is understable and in my opinion - can be potentially interesting for Water readers.
Authors attempted to present catastrophic floods in large river basins in many watersheds worldwide using a set of figures, tables and plots that are necessary and understable. The various parts of the manuscript are correctly written.
However, I have comments on the quality of the discussion: There should be more references to literature here, especially since the authors present this problem on a global scale. And here - we have references almost only related to rivers in Russia. I believe that this part should be slightly expanded with the remaining examples , especially those mendioned in the Introduction.
Author Response
Dear colleagues:
Actually I am a bit puzzled as the final version of the corrected article, uploaded into the Journal system (through the personal account), had no yellow marks (the article was uploaded both in PDF with line numbers and in Word + separate folder with all the drawings). Now, really yellow marks have appeared in the Manuscript for Revisions section. Probably, the Editors themselves have marked our corrections.
Further, the article contains 60 references to the works in this field and, surely, other authors, including official websites of the relevant departments of different countries, containing numerous links. We believe that it is quite enough for one article to understand its concept and content. At the same time, we are talking not only about the rivers in Russia. Though it is natural, as, firstly, many river basins of the world importance are situated here, which we (the authors from Russia) have studied. Secondly, we also use the data for the USA, Europe and India.
It seems quite representative for one article, which should still have a limited volume – it is not a global report of a FEMA-type agency, but a scientific sectorial article. We are currently preparing a new article concerning the catastrophic events of 2021-2022 in different countries containing a new detailed analysis. Perhaps we will send it to the same journal...
Thank you.
Prof. Tatiana Trifonova
April 05, 2022
02pm, GMT

Reviewer 2 Report
(1)The quality of graphics needs to be further improved, such as small font, low resolution problems.
(2)It is suggested to supplement the impact of sediment flow on international river drains, such as the Jinsha River basin in China.
(3)It is suggested to increase the literature on flood, sediment flow and their effects in Chinese studies.
(4)Some of the graphs lack basic information. Figure 2 does have the names of the abscissa and ordinate
(5)The article is too long, it is suggested to simplify part of the content.
Author Response
Dear colleagues:
(1) We did our best for making the drawings quality acceptable in the final version of the article, and especially in a separate folder with the drawings. It can be more detailed simply by increasing their format/size directly in the text of the article. However, if a respected Reviewer claims against a certain individualized illustrated material, but not against everything in general, we will respond.
(2) The Research concerning China (Jinsha River basin in China) is surely important, but it is a subject of separate consideration outside the scope of our article. Perhaps we will study it jointly with our Chinese colleagues, if they are interested…
(3) The article contains 60 links to works in this field and surely other authors, including the official websites of relevant departments of different countries, which include numerous links. We believe that it is quite enough for one article to understand its concept and content. At the same time, we are talking not only about the rivers in Russia. However it is natural, as firstly, many river basins of world importance are located here, which we (the authors from Russia) have studied. Secondly, we also provide the data for the USA, Europe and India.
It seems quite representative for one article, which still should have a limited volume – it is not a global report of a FEMA-type agency, but a scientific sectorial article. We are currently preparing a new article dealing with the catastrophic events of 2021-2022 in different countries with a new detailed analysis. Probably we will send it again to the same journal…
+ if item (2) is implemented, it will be done automatically.
(4) It is done for better understanding of the information directly from the figures for general presentation of the information without detailed reference to the text of the article directly.
(5) The presented volume of the article makes it possible to present the integrated material (including surface waters, groundwater, precipitation, landscapes, earthquakes – their analysis and modeling) self-sufficient for understanding and illustrating our concept in the selected type of manuscript as a Review.
+ if we take into account the Reviewer’s recommendations to paragraphs (2), (3), then the volume of the article will additionally increase. We‘d better make it in our new article devoted to the detailed specifics.
Thank you.
Prof. Tatiana Trifonova
April 05, 2022
02pm, GMT
